



# A one-year ACSM source analysis of organic aerosol particle contributions from anthropogenic sources after long-range transport at the TROPOS research station Melpitz

Samira Atabakhsh[1], Laurent Poulain[1], Gang Chen[2,3], Francesco Canonaco[2,4], André Prévôt[2], Mira Pöhlker[1], Alfred Wiedensohler[1], Hartmut Herrmann[1]

[1]Leibniz Institute for Tropospheric Research, Leipzig, 04318, Germany
[2]Laboratory of Atmospheric Chemistry, Paul Scherrer Institute, Villigen, Aargau, 5232, Switzerland
[3]MRC Centre for Environment and Health, Environmental Research Group, Imperial College London, London, W12 0BZ, U.K.
[4]Datalystica Ltd., Park innovAARE, Villigen, Aargau, 5234,Switzerland

*Correspondence to*: Hartmut Herrmann (herrmann@tropos.de)

## Abstract

Atmospheric aerosol particles are a complex combination of primary emitted sources (biogenic and anthropogenic) and secondary aerosol resulting from the aging processes such as condensation, coagulation, and cloud processing. To better understand their sources, investigations have been focused on source identification in urban areas in the past, while rural background stations are normally less impacted by surrounding anthropogenic sources. Therefore, they are predisposed for studying the impact of long-range transport of anthropogenic aerosols. Moreover, long-term measurements can help to study the potential temporal changes in the sources. Here, the chemical composition and organic aerosol sources of submicron aerosol particles were investigated at the Central European rural-background research station, Melpitz, using a one yearlong dataset determined by an aerosol chemical speciation monitor (ACSM) and a multi-angle absorption photometer (MAAP) from September 2016 to August 2017. Melpitz represents due to its location the Central European aerosol. It is an ideal location to investigate the impact of long-range transport, since the location is influenced by less polluted air masses from westerly directions and more polluted continental air masses from Eastern Europe. The organic aerosol (OA) dominated the submicron particle mass concentration and showed strong seasonal variability ranging from 39 % (in winter) to 58 % (in summer). It was followed by sulphate (15 % and 20 %) and nitrate (24 % and 11 %). The OA source identification was performed using rolling positive matrix factorisation (PMF) approach to account for the potential temporal changes in the source profile (SoFi Pro). It was possible to split OA into five-factors with a distinct temporal variability and mass spectral signature. Three were associated to anthropogenic primary OA (POA) sources: hydrocarbon-like OA (HOA, 5.2 % of OA mass in winter and 6.8 % in summer), biomass burning OA (BBOA, 10.6 % and 6.1 %) and coal combustion OA (CCOA, 23 % and 8.7 %). Another two are secondary/processed oxygenated OA (OOA) sources: less-oxidized OOA (LO-OOA, 28.4 % and 36.7 %) and more-oxidized OOA (MO-OOA, 32.8 % and 41.8 %). Since equivalent black carbon (eBC) was clearly associated with the identified POA



factors (sum of HOA, BBOA and CCOA, $R^2$= 0. 87), eBC's contribution to each of the POA factors was achieved using a
multi-linear regression model. Consequently, CCOA represented the main anthropogenic sources of carbonaceous aerosol
(sum of OA and eBC) not only during winter (56 % of POA in winter) but also in summer (13 % of POA in summer), followed
by BBOA (29 % and 69 % of POA in winter and summer, respectively) and HOA (15 % and 18 % of POA in winter and
summer, respectively). A seasonal air mass cluster analysis was used to understand the geographical origins of the different
aerosol types and show that during both winter and summer time, $PM_1$ (PM with aerodynamic diameter smaller than 1µm) air
masses with eastern influence was always associated with the highest mass concentration and the highest coal combustion
fraction. Since during winter time, CCOA is a combination of domestic heating and power plants emissions, the summer
contribution of CCOA emphasises the critical importance of coal power plants emissions to rural background aerosols and its
impact on air quality, through long-range transportation.
**1 Introduction**
Human health effects of air pollution from particulate matter (PM) are well known, and efforts are being made across the world
(*WHO, Expert Consultation*, 2019) to minimize both long-term exposure to harmful levels and air pollution peaks. Throughout
all the PMs, the submicronic particles known as $PM_1$ (particles with an aerodynamic diameter less than 1 µm), not only have
a negative impact on human health (Daellenbach et al., 2020) but also have a significant effect on visibility (Shi et al., 2014)
and climate (Shrivastava et al., 2017). Since the most numerous component of the atmospheric PM is the organic aerosol (OA)
(Jimenez et al., 2009; Chen et al., 2022), contributions to OA and explanations of its chemical and physical characteristics
remain challenging, whereas the large variety of OA can be attributed to primary emissions by various sources in different
seasons, as well as different reactions to atmospheric dynamics and complicated chemical mechanisms depending on
meteorological parameters and geographical locations.
In order to evaluate and recognize the sources of OA emission, aerosol mass spectrometers (AMS, Jayne et al., 2000) and
aerosol chemical speciation monitors (ACSM) (Ng et al., 2011; Fröhlich et al., 2013) are widely deployed worldwide (Chen
et al., 2022; Bressi et al., 2021; Fröhlich et al., 2015). AMS is commonly limited to short time periods due to the high
maintenance of the AMS measurements and their high operating costs. As a result, only a few studies run AMS continuously
(e.g., see Kumar et al., 2022 and O'Dowd et al., 2014). However, there was still a strong need for such a long-term analysis.
ACSM is designated for long-term monitoring purposes due to its robustness and much less labour-intense compared to AMS.
Therefore, the deployment of ACSM allows us to look at the long-term (more than one year) temporal changes and/or seasonal
variability of OA sources.
Regarding the identification of OA sources, source apportionment analysis using positive matrix factorisation algorithm (PMF,
Paatero and Tappert, 1994) was intensively used over the past two decades on both AMS and ACSM measurements (e.g. see
Crippa et al., 2014; Poulain et al., 2020). However, this algorithm faced two main limitations when used during a long time
period: firstly, the factor profiles are static over the analyzing period (Paatero, 1997); and secondly, rotational ambiguity which





provides non-unique solutions. To solve these issues, a multilinear engine (ME-2, Paatero, 1999) has been implemented in the
PMF analysis, which allows use of a priori knowledge to constrain the model to environmentally reasonable solutions (e.g.,
Lanz et al., 2008; Canonaco et al., 2013; Crippa et al., 2014). To consider the temporal variation of the factor profiles, a rolling
approach was suggested (Parworth et al., 2015; Canonaco et al., 2020). The rolling strategy involves advancing a smaller PMF
window (i.e., 14 days) and moving/rolling it over the whole dataset to catch the temporal changes of the source profiles with
a 1-day step.
Although several studies in Europe have already conducted source apportionment analyses of one year or more, most of them
were associated with urban or suburban environments (e.g., for urban studies: Stavroulas et al., 2019; Vlachou et al., 2019;
Huang et al., 2019; Qi et al., 2020; and for suburban studies: Katsanos et al., 2019; Y. Zhang et al., 2019), and only a few of
them were studied in rural-background sites (Schlag et al., 2016; Crippa et al., 2014; Vlachou et al., 2018; Paglione et al.,
2020; Dudoitis et al., 2016; Heikkinen et al., 2020; Chen et al., 2021; and Chen et al., 2022), although the rural-background
sites represent the major advantage to be able to study the impact of long-range transport of anthropogenic emissions and their
changes over a long time period. The Leibniz Institute for Tropospheric Research (TROPOS) Central European observatory
Melpitz has been continuously measuring aerosol chemical composition for 30 years. The station is a unique place in Europe,
sitting at the border between marine-influenced Western Europe and continental Eastern Europe. A direct consequence is that
the aerosol chemical composition and mass concentration strongly depend on the air mass origins, showing less polluted air
masses coming from the West and more polluted air masses from the East (Birmili et al., 2001; Spindler et al., 2010). However,
only a few studies were done on the source identification of the aerosol reaching the station by covering short time periods
mostly during winter (van Pinxteren et al., 2016, 2023).
The current study comprehensively investigates the $PM_1$ aerosol particle chemical compositions and the various OA sources
for Melpitz based on ACSM and multi-angle absorption photometer (MAAP) measurements from September 2016 to August
2017, using the most advanced rolling PMF with ME-2 implemented in the SoFi Pro package (Datalystica Ltd., Villigen,
Switzerland) (Parworth et al., 2015; Canonaco et al., 2013; Canonaco et al., 2020). Moreover, a multi-linear regression model
was used to estimate the contribution of equivalent black carbon (eBC) to the various PMF factors. Meanwhile, the influence
of air mass origin was investigated to identify the emission area of the different $PM_1$ sources.
**2 Methodology**
**2.1 Sampling site**
The atmospheric aerosol measurements were carried out at the TROPOS research station Melpitz (51.54° N, 12.93° E,
86 m a.s.l.), located approximately 50 km northeast of Leipzig, Germany. The station itself is mainly encircled by agronomical
pastures and forests within a rural area, which is why the station is recognized as a rural-background station (Spindler et al.,
2013). Since 1992, the station has been monitoring the influence of atmospheric long-range transport on background air quality
of Central European (e.g. Spindler et al., 2012 2013). The Melpitz station is part of EMEP (European Monitoring and



Evaluation Programme; Level 3 station, Aas et al., 2012), ACTRIS (Aerosol, Clouds and Trace gases Research Infrastructure),
GAW (Global Atmosphere Watch of the World Meteorological Organization), and GUAN (German Ultrafine Aerosol
Network, Birmili et al., 2009, 2015, 2016). For a general description of the chemical and physical aerosol characterization
analysis techniques, check e.g. Spindler et al., (2004, 2010, 2012, 2013); and Poulain et al., (2011, 2014, 2020).
**2.2 ACSM**
The chemical compositions and mass loadings of non-refractory $PM_1$ (NR-$PM_1$: organic, sulphate, nitrate, ammonium, and
chloride) with a 30-minute time resolution were measured by an Aerodyne quadrupole ACSM. The ACSM sampling technique
and operational information were previously detailed by Ng et al., (2011).
Briefly, after $PM_1$ transmits across a 100 μm critical orifice, the aerosols are centralized into a slender beam in an aerodynamic
lens (Liu et al., 2007). Non-refractory particulate material that evaporates at the oven temperature (generally 600°C) is recorded
and chemically determined using electron impact quadrupole mass spectrometry at 70 eV (Ng et al 2011). The ions are then
detected using a quadrupole residual gas analyser (RGA, Pfeiffer Vacuum Prisma Plus). The ACSM takes 30 second samples
of both ambient and particle-free air. The difference in these measurements identifies the aerosol mass spectrum. To change
the signal spectra into organic or inorganic species concentrations, the fragmentation table (Allan et al., 2004), the ion
transmission correction, and the Response Factor (RF) are applied. To improve the particle loss as a result of bouncing off the
vaporizer, the ACSM data were processed according to manufacturer guidelines with using a composition dependent collection
efficiency (CDCE) correction relying on the algorithms suggested by Middlebrook et al. (2012). Calibrations of Ionization
Efficiency (IE) and Relative Ion Efficiency (RIE) were performed using a 350 nm monodispersed ammonium nitrate and
ammonium sulphate (Ng et al., 2011). The final mean value for IE was $4.93(\pm1.45) \times 10^{-11}$ and the mean values for RIEs for
ammonium and sulphate respectively were 6.48±1.26, and 0.68±0.13. Details on the QA/QC for this dataset can be found in
Poulain, et al., (2020).
**2.3 Additional measurements**
In parallel to the ACSM, a MAAP was used to measure the mass concentrations of equivalent black carbon (eBC) (model
5012 Thermo Scientific; Petzold and Schönlinner, 2004). Conversion of the eBC mass concentration from the $PM_{10}$ inlet to
the ACSM $PM_1$ cut-off was made by applying a correction factor of 0.9 following Poulain et al (2011). Furthermore, a dual
mobility particle size spectrometer (TROPOS-type T-MPSS; Birmili et al., 1999) was used to measure the PNSD from 3 to
800 nm (mobility diameter, d mob) at ambient and 300°C temperatures (Wehner et al., 2002). The MAAP was situated in the
same laboratory container as the ACSM and these instruments sampled the same $PM_{10}$ inlet after a dryer, and the sampled air
distribution among the instruments was equally assured by an isokinetic splitter (Poulain et al., 2020).
In addition to the online measurements, high-volume samplers (DIGITEL DHA-80, Digitel Elektronik AG, Hegnau,
Switzerland) were utilized to capture daily $PM_{2.5}$ samples on a quartz filter (for 24 hours from midnight to midnight). For more





details on the sample preparation and evaluation methods, see Spindler et al., (2013). Levoglucosan as a tracer for wood
burning combustion was measured following Iinuma et al., (2009) using high performance anion exchange chromatography
coupled with an electrochemical detector (HPAEC-PAD) that was used for the analysis of anhydro-monosaccharides (Iinuma
et al., 2009).
Trace gas measurements were also carried out. Ozone was determined by a U.V. Photometric gas analyser mode 49C (Thermo
Scientific, UK), $SO_2$ by an APSA-360A (Horiba, Kyoto, Japan), and NO and $NO_2$ using a customized Trace Level NOx
Analysis Model 42i-TL (Thermo Scientific) equipped with a blue light converter. Standard meteorological parameters
(temperature, relative humidity, solar radiation, precipitation, wind direction, and wind speed) were regularly measured.

**2.4 Rolling PMF (ME-2) source apportionment of OA**

This work conducted the most advanced source apportionment analysis following a standardized protocol developed by Chen
et al., (2022). The PMF method was used to allocate the source of the OA (Paatero and Tappert, 1994) through the Source
Finder professional (SoFi Pro, Canonaco et al., 2021) software package (Datalystica Ltd., Villigen, Switzerland), within the
Igor Pro software environment (Wavemetrics, Inc., Lake Oswego, OR, USA). Two matrices of factor profiles *F* and factor
contributions *G*, defined the dataset *X*, and the matrix *E* named the residual matrix is the fraction which cannot be described
by the model. Time series and the chemical fingerprint of sources respectively have been represented by $F_{kj}$ and $G_{ik}$,
respectively. The dimension of $F_{kj}$ and $G_{ik}$ are based on the order *p*, which is the number of factors selected to represent the
data which is defined by the user:
$X_{ij} = \sum_{k=1}^{p} G_{ik} \times F_{kj} + E_{ij}$ (1)
In this study, since the measurement covers a period of 12 months (full four seasons), four separate PMF inputs were prepared.
Unconstrained PMF was applied with 4 to 6 factors runs for all the seasons; throughout the pre-result and while referring to
previous studies (Crippa et al., 2014 and van Pinxteren et al., 2016) primary factors were separated as hydrocarbon-like OA
(HOA), biomass burning OA (BBOA) and coal combustion OA (CCOA). However, unconstrained PMF did not result to
separate the primary factor profiles. Introducing constraints based on prior knowledge is an efficient strategy for avoiding the
mixing of primary factors (Canonaco et al., 2013; Crippa et al., 2014). For this reason, the multilinear engine (ME-2) algorithm
(Paatero, 1999) enables the incorporation of time series and factor profiles constraints in form of the *a*-value approach. In
dealing with a profile constraint, the *a*-value specifies the variety of a factor that can deviate from the anchor profile during
the PMF iteration:
$f_{j,solution} = f_j \pm a \cdot f_j$ (2)





The constraints applied through ME-2 for HOA and BBOA sources used the anchor profile of Crippa et al., (2014), and Ng et
al., (2010), respectively. The anchor profile used for CCOA was generated from our own winter data during this work (SI,
1.1). For each of the four seasons, primary profiles were subject to a sensitivity analysis with *a-values* ranging from 0-0.4 for
HOA and BBOA, 0-0.5 for CCOA, and steps of 0.1 to choose the best *a-value* combination for these three factors.
In the PMF approach, there is the intrinsic property of static factor profiles during the period of PMF analysis. Even though
for short-term measurements (like one/two season/s) this might be a sensible estimation, long-term observations as are typical
for current ACSM study (one year and more), are expected to be subject to evolving factor profiles based on seasonality. To
consider the temporal changes, the rolling PMF window method was developed (Canonaco et al., 2021b; Parworth et al., 2015).
This technique is applied to a small window, which is slowly extended throughout the whole dataset. Based on the dataset, the
user determines the width of the PMF window, the shift parameter, and the number of PMF repeats per window; for the current
work, we set 14-day windows, 1-day shifts, and 100 repeats per window.
In addition, this rolling PMF analysis was coupled with the bootstrap re-sampling approach (*Bootstrap Methods: Another Look*
*at the Jackknife on JSTOR*, 1979), which can randomly select a part of the original matrix and repeat a part of the rows to
generate a new same-sized matrix to test the stability of solutions and to estimate the statistical error. Overall, we have
combined rolling PMF with ME-2 and bootstrap to conduct the source apportionment investigation, and more information on
this new approach was described in Canonaco et al., (2020). This approach for a yearlong dataset generates an enormous
amount of PMF runs (N= 35800) and not all of the solutions are environmentally reasonable. Since it is practically impossible
to manually inspect all PMF runs, the criterial-base selection was introduced in SoFi Pro to automatically and objectively
select environmentally reasonable PMF solutions (Canonaco et al., 2020). Finally, the resulting factors were interpreted as
HOA, BBOA, CCOA and two oxidized OA (OOA) factors named less-oxygenated OOA (LO-OOA) and more oxygenated
OOA (MO-OOA). The steps and setups utilized in the evaluation of this dataset are detailed in the supplement (Sect. 1).
**2.5 Air mass trajectory analysis**
Non-parametric wind regressions (NWR) were used to approximate the OA source concentrations at a given wind direction
and speed (Henry et al., 2009). The NOAA HYbrid Single-Particle Lagrangian Integrated Trajectory (HYSPLIT-4) model was
used to analyse 96 h backward trajectories at 500 m above the model ground of the sampling place (Draxler and Hess, 2004).
The trajectory results were used for two independent but complementary analyses to better depict the emission area of the
aerosol: by identifying the potential aerosol sources area and by clustering the trajectories.
A cluster analysis of the different trajectories was performed. The synoptic-scale air mass condition, together with geographical
location and paths, is a crucial driver of local pollutant concentrations (e.g. Sun et al., 2020; Ma et al., 2014). Local particle
mass concentrations and meteorological conditions can play a significant role and be associated with specific air mass
trajectories. In addition, the trajectories of the air mass can influence aerosol compositions. For example, the stability of the
atmosphere is also meaningful since it influences both the vertical dilution of pollutants and the overall particle mass



concentration. Therefore, the effects of inter-annual variations in air mass conditions and the stability of atmosphere on observed patterns were inspected using a self-developed back-trajectory cluster method (BCLM), concerning air mass backward trajectories, pseudo-potential temperature profiles, $PM_{10}$ mass concentration profiles over Melpitz, and seasons (Birmili et al., 2010; Ma et al., 2014). Descriptive analysis, cluster processing, and data processes and products are all described in detail by Sun et al., (2020) and Ma et al., (2014).

## 3 Results

### 3.1 $PM_1$ chemical composition

In this work, we investigate one-year long measurements of $PM_1$ for Melpitz, Germany. All the data is presented in UTC, during the winter and summer, the time zone is one and two hours behind local time, respectively. Yearly time series, seasonal variation, and diurnal cycles of aerosol particle chemical compositions including mass concentration and mass fraction, as measured by ACSM and MAAP, are shown in figures 1, 2, and 3, respectively. Over the entire period, the chemical composition of $PM_1$ was basically made up of organic aerosol (46 % of the total mass; Fig. 1c), sulphate (16 %), nitrate (21 %), ammonium (11 %), eBC (6 %), and chloride (close to 0 %). However, a mean mass concentration of 10.47 µg/m³ (Fig. 1) was obtained with an obvious seasonal trend which detected the highest total mass concentrations (15.95 µg/m³) during the winter time and lowest mass concentration during the summer time; 6.24 µg/m³ (Fig. 1a and Fig. 2a). Compared to previous AMS measurements of Poulain et al., (2011) at the same station, a similar seasonal trend was observed in the period 2008/2009, while the absolute masses differed (Table. S1), which is at least partially related to the inter-annual changes of the meteorological conditions. Fig. S2 presents the coming high polluted air masses for total $PM_1$ to the measurement site in the current study; the polluted Eastern Europe flow with high mass concentration and south-west with low mass concentration was more clearly found in winter time rather than in other seasons, which will be comprehensively discussed in the Sect. 3.4.

In comparison with other ACSM/AMS rural-background stations in Europe which can be divided into three parts Northern Europe (NE), Southern Europe (SE), and Mid-latitude Europe (ME) (Bressi et al., 2021), the annual $PM_1$ mean mass concentration measured at Melpitz is similar to the value obtained at other ME stations, such as Magadino 10.1 µg/m³, Kosetice 8.5 µg/m³ ( Chen et al., 2022), 9.1 µg/m³ on average of $PM_1$ mean mass concentration of 6 stations (Ispra, Melpitz, Magadino, Cabauw, Sirta and Hohenpeissenberg, Bressi et al., 2021).

### 3.1.1 Inorganic

The seasonality of the inorganic species can be associated with their variations in emissions and/or the changes in their chemical atmospheric processes. Throughout the year, the mass concentration and their respective contribution to the total $PM_1$ mass of nitrate, ammonium, and chloride increased from a minimum value in summer (11 %, 7 %, and 0 %, respectively; Fig. 2b) and reached a maximum value in winter (24 %, 12 %, and 1 %, respectively; Fig. 2b). Sulphate showed a slightly different behavior. Although the contribution of sulphate to the total $PM_1$ decreased slightly from summer (20 %) to winter



(15 %), its mass concentration remained higher in winter compared to summer (2.38 µg/m$^3$ and 1.23 µg/m$^3$, respectively;
Table. 1). The enhancement is not as drastic as other inorganic species since sulphate is least volatile, therefore, more fraction
of sulphate stayed in particle phase even in summer. Moreover, with enhanced irradiations in summer, sulphate formation from
photochemistry could be enhanced as well. This result is consistent with the mean PM$_1$ mass concentration measured by AMS
for the three periods during fall (16. September.2008 to 03. November.2008), winter (24. February.2009 to 25. March.2009),
and summer (23. May.2009 to 09. June.2009) campaigns reported by Poulain et al., (2011). The diurnal cycles of sulphate
(Fig. 3) showed a different daily pattern in warm and cold seasons. In summer, sulphate mass concentration increased during
the day and reached its maximum level at 12:00 UTC (Fig. 3) due to sulphur dioxide photochemical oxidation processes in the
atmosphere, which also presented the highest mass concentration during the day, along with maximum temperature and sun
radiation in summer time (Fig. S3). Furthermore, the wind rose analysis showed a high mass concentration of sulphate at low
wind speed (Fig S3). Although locally formed emissions of sulphate (Fig. S2) can explain this peak during the day in summer,
this photochemical process is not the only source of sulphate. It especially cannot explain the highest mass concentrations
during the winter time with almost no diurnal variation (Fig. 3). For winter, the emission of domestic heating processes, which
could be enhanced in the atmospheric boundary layer (Stieger et al., 2018), along with the long range transported emissions,
which came from north-east toward the measurement site (Fig. S2), and also high ammonium nitrate due to partitioning
according to temperature, explain the high mass concentration but the low relative contribution of sulphate.

Nitrate is mostly found in the form of ammonium-nitrate (NH$_4$NO$_3$), which is reliant on the gas phase precursor concentrations,
temperature, humidity, and aerosol chemical composition (Poulain et al. 2011; Stieger et al., 2018). Both nitrate and
ammonium showed a minimum mass fraction and mass concentration in summer (11 %, 0.68 µg/m$^3$, 7 %, 0.43 µg/m$^3$,
respectively; Fig. 2), an increasing trend toward the cold months and reached their maximum mass fraction and mass
concentration in winter time (nitrate 24 %, 3.87 µg/m$^3$, ammonium 12 %, 2 µg/m$^3$, respectively; Fig. 2). The diurnal cycles of
nitrate and ammonium (Fig. 3) showed a relatively similar daily pattern in all seasons, which means the highest values were
reached in the morning, due to the beginning of vertical mixing and a reduction in the afternoon followed by an increase during
the night, reflecting their night time production during every season. The volatile behaviour of ammonium-nitrate strongly
affects its temporal variation during warm days leading to the formation of the gaseous nitric acid and ammonia compounds
at higher temperatures and low humidity (Fig. S3, and S3). In winter, ammonium-nitrate remains mainly in the particle phase
(Seinfeld and Pandis, 2006) and, like sulphate, arrived at the measurement site due to the long-range transported emissions
which not only came from the north-eastern but also south-western flow, describing higher mass concentrations for nitrate and
ammonium (Fig. S2). High values of nitrate and ammonium in spring time are linked to agronomical fertilization (Stieger et
al., 2018). These seasonal contribution results for both, nitrate and ammonium, are consistent with the previous AMS study
(Poulain et al., 2011), with minimum fraction to the total AMS-PM$_1$ during summer (nitrate 5 % and ammonium 8 %; Table.
S1), and maximum fraction during winter time (nitrate 34 % and ammonium 17 %; Table. S1). However, it is known that a





fraction of the nitrate signal can be attributed to nitrogen containing organic species (Kiendler-Scharr et al., 2016), which can
affect the overall nitrate mass concentration (Poulain et al., 2020).

Although chloride had the lowest annual mass concentration (0.05 µg/m$^3$) compared to all other PM$_1$ chemical components
(Table. 1), it showed the highest mass concentration and mass fraction in winter (0.11 µg/m$^3$, 1 %, respectively; Fig. 2a&b;
Table. 1) compared to the other seasons; as seen in the previous AMS study of Poulain et al., (2011) (2 %, Table. S1). It could
be related to the surrounding and transported emissions which were high for air masses from north-easterly and south-westerly
directions (Fig. S2). In a multi-year analysis of the hourly PM$_{10}$ chloride mass concentration measurements using a MARGA,
Stieger et al., (2018) attributed the chloride sources of Melpitz during winter to the resuspension of road salt used for the de-
icing of streets, mainly coming from the cities of Torgau and Leipzig. These sites are also located in the wind directions along
with the coal and wood combustion emission region, which could explain the highest mass concentration of chloride during
the winter. Furthermore, the existence of chloride might be due to low mass concentration marine influences consisting of sea-
salt aerosol during all the seasons in the south-westerly direction (Fig. S2) which was previously studied by Stieger et al.,
(2018). However, it is known that the AMS-technology cannot properly detect sea salt (S. Huang et al., 2018; Ovadnevaite et
al., 2014) because the majority of chloride is in the refractory part which cannot be flash vaporized at 600 ℃. Consequently,
the chloride detected by the ACSM is mostly related to combustion processes (wood, coal combustion as well as trash burning;
Li et al., 2012).
**3.1.2 eBC and organics**
The eBC showed its maximum mass concentration and mass fraction to PM mass during winter time at 1.38 µg/m$^3$ and 9 %,
respectively (Fig. 2), and only 0.25 µg/m$^3$ and 4 %, respectively, during summer time (Fig. 2). This is consistent with the
expected highest anthropogenic emissions from fossil fuel consumption (house heating and energy productions) in winter
compared to summer (Spindler et al., 2010). Furthermore, considering measured eBC in regard to wind speed and wind
direction (Fig. S2), the highest mass concentrations could be linked to north-easterly and south-westerly winds for fall, winter,
and spring seasons, while in summer time it is mostly linked to the surrounding emissions (Fig. S2). Significant changes in the
diurnal profiles of eBC for the different seasons can be found with the highest mass concentrations throughout the cold months
compared to warm months owing to house heating (Fig. 3). It also showed morning and evening peaks during all seasons (Fig.
3). This is consistent with those observed for the nitrogen oxides (Fig. S3), which might be attributed to liquid fuel emissions
and possibly the impact of the traffic rush hours on the main street, B 87, located approximately 1 or 1.5 km north of the
station, (Yuan et al., 2021). In the following chapter, diurnal patterns showed lower mass concentrations at noon, and increased
in the late afternoon to become nearly constant from 8 p.m. until midnight (Fig. 3). This ambient particulate pollution resulting
from very surrounding sources in the village was reported by van Pinxteren et al., (2023). Diurnal increments of eBC were
smaller in fall and spring compared to winter; the increment in summer is also correspondingly low due to the absence of
house heating emissions, and the diurnal variation in the increment is determined by surrounding motor vehicle emissions in





combination with the mixing layer height (van Pinxteren et al., 2023). Further discussions on the seasonal trend of the eBC
can be found in Sect. 3.3.

Organic aerosol (OA) was the predominant species throughout the whole year, with a mean mass concentration of 4.84 µg/m³
and a mass fraction of 46 % (Fig. 1c; Table. 1). The OA mass fraction decreased from the maximum value in summer and
attained a minimum mass fraction in winter (58 %, 39 %, respectively; Fig. 2b). Similar to the comparison of previous inorganic
AMS measurements performed at Melpitz (Poulain et al., 2011), AMS-OA contribution to total PM₁ showed maximum
contribution during summer (59 %, Table. S1), and minimum contribution during winter (23 %) as well. However, the mass
concentration of OA increased from its lowest value in summer and reached its highest value in winter time (3.67 µg/m³, 6.21
µg/m³ respectively; Fig. 2, Table. 1). Similar to eBC, OA measured in according to wind direction and wind speed showed
highest average mass concentrations for north-easterly and south-westerly winds in winter (Fig. S2). In fall, polluted air masses
came from the north-easterly direction, and in spring and summer OA, surrounding emissions closer to Melpitz were identified
(Fig. S2). The diurnal cycle of the organic had an identical pattern across all seasons (Fig. 3), showing the highest mass
concentration night time, a small peak in the early hours of the morning related to rush hours, and the lowest mass
concentrations around the early afternoon. The peak observed around 12:00 UTC in summer time (Fig.3) can be due to the
local photochemical production that leads to the formation of secondary organic aerosol mass during the day, similar to the
diurnal behavior of sulphate (previously discussed in Sect. 3.1.1). However, the reduction in total OA mass concentration
throughout the day (Fig. 3), which was mostly observed during the warm seasons (spring and summer), could be clearly related
to the dilution effect of increasing mixed layer height.

Overall, eBC and OA can be composed of various sources with strong seasonal dependencies, as well as be influenced by
different responses to atmospheric dynamics depending on meteorological parameters, geographical locations, and chemical
processes. Therefore, a comprehensive analysis of the OA and eBC sources was performed using source apportionment
techniques.
**3.2 Source apportionment of OA**
The chosen solution for the organic aerosol source apportionment contained five different factors based on their time series
and mass spectra (Fig. 4). The source apportionment solution is based on a partly constrained rolling approach with three
primary organic factors (POA), namely HOA (on average 0.30 µg/m³ and 6 % of the total OA; Table.1 and Fig. 4), BBOA (on
average 0.39 µg/m³ and 7.9 % of the total OA) and CCOA (on average 0.77 µg/m³ and 15.4 % of the total OA). In addition to
these POA factors, two oxygenated organic aerosols (OOAs) were identified as LO-OOA (on average 1.62 µg/m³ and 32.4 %
of the total OA), and MO-OOA (on average 1.92 µg/m³ and 38.4 % of the OA). The seasonal average mass concentrations and
relative mass fractions of each OA factor to the total OA mass and their seasonal diurnal variation are presented in Figures 5
and 6; respectively. They will be discussed separately in the following sections.





### 3.2.1 POA factors

The HOA mass spectrum (Fig. 4b) is recognized by mass fragments at unsaturated and saturated hydrocarbon chain pairs $m/z$ 41 ($C_3H_5$), 43 ($C_3H_7$), $m/z$ 55 ($C_4H_7$) and 57 ($C_4H_9$) (Zhang et al., 2005; Canagaratna et al., 2004), which are representative of liquid fuel combustion emissions and are associated with either traffic emissions or domestic heating fuel (Wang et al., 2020). This result designates HOA as a minimal source of OA at the monitoring site, which is consistent with previous studies in the $PM_1$ range made in the same place: a) total average was 7 % of the organic mass concentration in a study by Crippa et al., (2014) total average was 3 % of PM size range between 0.05-1.2 µm mass concentration in a study by van Pinxteren et al., (2016) (Table. S2). However, in comparison with other ACSM/AMS stations in Europe (22 stations; Chen et al., 2022), Kosetice with 9.7 % as a rural-background site, and Bucharest with 13.7 % as an urban-background site showed the minimum annual HOA mean contribution of total OA, which is similar to the contribution at Melpitz.

Mass concentration of HOA followed a slightly increasing seasonal pattern towards the cold months, from 0.23 µg/m³ in summer to 0.36 µg/m³ in the winter (Fig. 5a; Table. 1). HOA presented a low correlation with nitrogen oxides over the entire period ($R^2$= 0.17, Table. 1), but it correlated well with eBC in winter ($R^2$= 0.52; Table. 1) and shows a weaker correlation in summer ($R^2$= 0.28; Table. 1). Possibly HOA is also associated with household heating (35 % by oil and 11 % by liquid petroleum gas, van Pinxteren et al., 2023) rather than traffic emissions, especially during the cold months. Analyzing the pollution wind rose, the highest winter HOA mass concentrations are associated with the north-easterly wind direction regardless of the wind speed suggesting the influence of long-range transported emissions (Fig. 7). During the warm months, the emissions were more from the surrounding area, still associated with a north-easterly wind direction but only at low wind speed (Fig. 7), which might be associated with either the surrounding traffic emissions, as well as the domestic emissions associated not with house heating in summer but with hot water production (van Pinxteren et al., 2023).

The diurnal patterns of HOA reproduced two peaks in the morning and evening for all seasons (Fig. 6). The small time shift for the start of the evening increase corresponds to the time shift of the sunrise between winter and summer. The diurnal cycles reached a systematic minimum during the day time probably not only owing to emission decrease but also emphasizing the effect of dynamic atmospheric processes (e.g. mixing layer height (MLH) and planetary boundary layer (PBL)) (Fig. 6, and S4). Oppositely to what can be seen during the day time, night time mass concentrations appeared to be unaffected by the seasons, showing similar mass concentrations all year round, i.e. their mass concentration rose continuously in the early evening and remained at a very similar mass concentration over the night, which supports the hypothesis of yearlong continuous rather surrounding emissions.

The mass spectra of BBOA are identified by ions at $m/z$ 29, 43, 60, and 73 (Fig. 4b), known as fragments tracers of anhydro-sugars like levoglucosan (Alfarra et al., 2007), which have been identified as indicators of wood combustion processes (Simoneit et al., 1999; Simoneit and Elias, 2001). This is confirmed by the correlation between BBOA and levoglucosan over the whole period ($R^2$= 0.65; Table. 1). On average, BBOA mass concentration and contribution were 0.39 µg/m³ and 7.9 %,





respectively (Table. 1 and Fig. 4a). However, its contribution is highest during winter time (10.6 %; Fig. 5), which is similar
to previous studies in different PM ranges for the Melpitz station during the cold months: a) in $PM_1$ range, 14 % of OA mass
concentration in fall (Crippa et al., 2014); b) in 0.05-1.2 µm range, highest contribution with 10 % of PM mass concentration
in winter (van Pinxteren et al., 2016); and c) in $PM_{10}$ range, highest contribution with 16 % of PM mass concentration in winter
(van Pinxteren et al., 2023).
The high value of BBOA is mainly attributed to residential heating, and indicates the effect of transported biomass burning
emissions to the sampling site in cold months (Fig. 7), while in summer time, it is still observable as surrounding emissions
during periods of low wind speed (Fig. 7 and Fig. S4) with a mass concentration of 0.21 µg/m³ and a contribution of 6.1 % to
total OA (Fig. 5). The presence of BBOA in the summer can be linked to water heating systems using wood briquettes and
logs (estimated at 32 % of total central heating in this area, van Pinxteren et al., 2023). Moreover, it can also be related to
recreational open fires and/or barbecue activities (van Pinxteren et al., 2023). This result is similar to other ACSM/AMS rural-
background stations in Europe (22 stations; Chen et al., 2022); both Magadino and Kosetice showed the highest contribution
of BBOA during winter time (27.4 % and 15.5 % respectively).
The diurnal cycles, peaking from early evening to early morning in winter (Fig. 6), match the expectations for a factor related
to domestic heating activities, along with a better eBC correlation during winter than during summer time ($R^2$= 0.81, and $R^2$=
0.42, respectively; Table. 1). Finally, in opposition to HOA, the night time BBOA mass concentration showed a strong seasonal
variation having its highest mass concentration during winter nights and lowest during summer time, the influence of the
impact of house heating emissions on the BBOA emissions. However, the day time behavior reflects the influence of enhanced
vertical mixing during day time (higher temperature, Fig. S3) combined with high wind speeds (Fig. 11) can readily cause
dilution and thus low pollutant concentrations near the ground (Chen et al., 2021; Via et al., 2020; Paglione et al., 2020).

The mass spectrum of CCOA is characterized by fragments at *m/z* 77, 91, and 115 (Fig. 4b) as previously reported by Dall'Osto
et al., (2013); Xu et al., (2020); Tobler et al., (2021) and Chen et al., (2022). These specific fragments can be associated with
unsaturated hydrocarbons, particularly ion peaks related to polycyclic aromatic hydrocarbon (PAH). The CCOA time series
showed the strongest correlation with eBC ($R^2$= 0.9; Table. 1). In addition, several studies reported that coal combustion
emissions are often accompanied by high chloride mass concentration (e.g; Iapalucci et al., 1969; Yudovich and Ketris, 2006
and Tobler et al., 2021). Here, the correlation between CCOA and chloride was higher during winter than during summer time
($R^2$= 0.41, 0.15 respectively; Table. 1), as the gas-particle phase equilibrium dramatically changes with rising temperatures
(Tobler et al., 2021). Although chloride is almost observable in the particle phase as ammonium chloride ($NH_4Cl$) at lower
temperatures, chloride is typically observable in the gas phase as hydrogen chloride (HCl) at higher temperatures (Tobler et
al., 2021).
CCOA represented on average 15.4 % of the total OA (0.77 µg m⁻³), (Table. 1; Fig. 4a) and is the most important POA over
the entire period. No CCOA factor was identified in the previous AMS measurements made at Melpitz (Crippa et al., 2014).
Most likely this factor was not properly resolved and/or it was not possible to properly separate it from the other factors since





no reference mass spectra for CCOA was reported in the literature at that time. CCOA showed the highest mass concentration
and mass fraction during the winter (1.58 µg/m³, 23 %, respectively; Fig. 5a; Table. 1), which is related to the surrounding
emissions and long-range transported air masses coming from two different directions, north-easterly and south-westerly (Fig.
7). Not surprisingly, the lowest mass concentration and contribution were observed during the summer time (0.30 µg/m³, 8.7
%, respectively; Fig. 5a; Table. 1,) which most probably correspond to only long-range transport as later discussed in Sect. 3.4
(Fig. 9). Moreover, this result is consistent with previous measurements made in the same place. For the size range 0.05-1.2
µm van Pinxteren et al., (2016) reported a contribution of 29 % and 21 % of the PM in winter and summer respectively, and a
contribution of 7 % and 0 % for winter and summer respectively for the $PM_{10}$ range was found (van Pinxteren et al., 2023).
From all ASCM/AMS stations (22 stations; Chen et al., 2022) only Melpitz as a rural-background site and Krakow as an urban-
background site showed the coal combustion emissions with the maximum contribution during winter for both sites (Krakow:
18.2 % and Melpitz: 23 %) compared to summer (Krakow: 4.5 % and Melpitz: 8.7%). The drastic seasonal changes in Krakow
are attributed to the common use of coal burning for residential heating reasons during the winter time (Casotto et al., 2022;
Tobler et al., 2021), while in Melpitz, as discussed above, coal combustion is affected by both surrounding and transported
emissions from other sites.
Mass concentrations of CCOA during night time were much higher than during day time throughout all seasons (Fig. 6), further
verifying the increased coal combustion emissions from coal heat generation at winter time night and the potential decrease in
emissions during the day due to a strong influence of atmospheric dynamics.

**3.2.2 OOA factors**

The two OOAs (Fig. 4) referred to as LO-OOA and MO-OOA are known to be characterized by the different ratios of their
$m/z$ 43 and $m/z$ 44 fragments (Fig. 4b), that represent the oxidation level (Canagaratna et al., 2015). While $m/z$ 43 could be
derived from $C_2H_3O^+$ (a signature of the semi-volatile) and/or $C_3H7^+$ (a signature of the primary emissions of the hydrocarbon-
like), $m/z$ 44 is mainly derived from the fragment of $CO_2^+$ (a signature of oxygenated, particularly acids) (Canonaco et al.,
2015; Ng et al., 2010). As presented in Fig. 4b, MO-OOA mass spectra showed a notable peak at $m/z$ 44. This spectrum has
been extensively recognized as low volatility OOA (LV-OOA) and described to be made up of aged secondary OA (SOA) and
highly oxidized OA ( Lanz et al., 2007; Ulbrich et al., 2009; Q. Zhang et al., 2011; Ng et al., 2011b); while the mass spectra
of LO-OOA in this study presented a higher $m/z$ 43 (Figs. 4b) compared to MO-OOA, which is similar to the mass spectral
pattern of the previously reported freshly formed semi volatile OOA (SV-OOA) (Jimenez et al., 2009; Ng et al., 2010). To
differentiate the variations of OOAs factor, the $f44$ vs $f43$ space was used which is a typical diagnostic tool based on
atmospheric aging (Ng et al., 2010).
The seasonal $f44/f43$ for OOAs measured points and the $f44/f43$ for modelled factor profiles (LO-OOA and MO-OOA) are
presented in Fig. S4. The data points in Fig. S4 are distributed differently according to the season (Chen et al., 2021; Canonaco
et al., 2015; Crippa et al., 2014; Chazeau et al., 2022). Furthermore, the modelled factor profile points represent a high
variability in space, especially for LO-OOA. This assumes how an annual or seasonal PMF solution, unless a larger number





of factors are used, would perform poorly in capturing all of the variations of SOA. In order to capture time-dependent changes,
in particular for LO-OOA, it is, therefore, advantageous to perform rolling PMF analysis. The triangle plot defined by Ng et
al. (2010) is also shown in Fig. S4. As assumed the LO-OOA points were concentrated in the lower part of the space, whereas
more aged MO-OOA points relocated to the upper part of the space during the aging process. The fall, spring, and summer
data points were all located on the right side of the triangle (Fig. S4), however the winter data points were located near the top
and inside the triangle. The data points on the right side of the triangle correspond to the time exposed to higher temperatures
more than those that are within the triangle. This could be attributed to an increase in biogenic SOA emissions if the temperature
increased, as biogenic OOA appears to be dispersed all along right the side of the triangle. Further, as the temperature is
reduced, the increased biomass emissions cause the OOA points to lie vertically inside the triangle, as seen in the winter data.

The two OOAs were the two most significant contributors to the total OA fraction (Fig. 4) over the entire period. The seasonal
mean mass concentrations of MO-OOA varied from higher mass concentrations during winter (2.25 µg/m$^3$) and lower during
summer time (1.44 µg/m$^3$, Table. 1). However, the highest MO-OOA mass concentrations found during the cold periods are
similar to the seasonal patterns in POA. Furthermore, high mass concentrations of MO-OOA are generally found at high
relative humidity (RH > 80 %) and low temperature (< 0 ℃), i.e., conditions during winter time (Fig. S5). This low air
temperature condition can be linked to a possible scenario for an increase in the MO-OOA precursor emissions from biomass
burning and coal combustion as a result of residential heating activities during winter time. Therefore, significant enhancement
appears to be an effect of RH during winter, proposing that the aqueous-phase heterogeneous mechanisms could also play a
crucial way in the regional MO-OOA formation through winter as suggested by Gilardoni et al., (2016). In contrast, no RH-
temperature-dependent trends for the MO-OOA were found in the other seasons (Fig. S5), indicating more complex formation
processes during other seasons. Meanwhile, MO-OOA diurnal cycles presented a seasonal variation as well, with a remarkable
enhancement in the evening and night time during winter (Fig. 6), indicating a potential regional formation mechanism
containing night time chemistry (Tiitta et al., 2016). While in fall, spring and summer, MO-OOA displayed a considerable
increase during the day (Fig. 6), indicating that higher temperatures result in considerable regional photochemical production
of SOA particles (Fig. S3) and enhanced solar radiation (Petit et al., 2015). Furthermore, regarding the correlation of mass
concentration of MO-OOA with sulphate, the latter is regarded as a local secondary production indicator (Petit et al., 2015,
and Table. 1). Consequently, alongside almost stable mass spectra throughout the year, MO-OOA seems to be derived from a
variety of seasonal-dependent formation mechanisms and sources (such as aged background, biomass burning, coal
combustion, and biogenic sources).

The seasonal mean mass concentrations of LO-OOA varied from higher mass concentrations during fall (2.13 µg/m$^3$) and
lower mass concentrations during spring time (1.24 µg/m$^3$, Table. 1). Temperature had a significant effect on LO-OOA, and
showed a distinguishable seasonal variation pattern. The temperature-RH dependence of the LO-OOA was not quite similar
depending on the season (Fig. S5). The highest winter time LO-OOA mass concentrations were found mostly at low

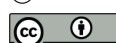



temperatures and high RH environments, indicating that gas-particle partitioning might have a key role in LO-OOA formation
throughout this season. The freshly formed SOA deriving from primary biomass burning and coal combustion emissions, as
found in previous studies (Crippa et al., 2013; Zhang et al., 2015; Y. Sun et al., 2018; Stavroulas et al., 2019) can also affect
the LO-OOA during the cold months. Furthermore, during winter time the correlations between LO-OOA and nitrate ($R^2$=
0.59) were found. Different LO-OOA daily cycles were also found in different seasons (Fig. 6). The daily changes in LO-
OOA displayed higher mass concentrations in night time compared to day time in fall, spring, and summer (Fig. 6), highlighting
the significant roles of night time chemistry and/or gas-particle partitioning in the LO-OOA formation, while the decrease
during the day is partly linked to the atmospheric dilution effect (Fig. S3), evaporation and photochemical aging into MO-
OOA (Fig. 6). For winter night increments, lower temperature in favor of condensation; and more abundant precursors present
considering increased BBOA emission, therefore enhanced night chemistry activities, leads to higher LO-OOA; moreover,
shallow boundary layer in winter and night time inversion caused pollutants to accumulate.
**3.3 eBC source apportionment**
The eBC correlated with each of the three identified primary organic factors (HOA, BBOA, and CCOA) during the source
apportionment analysis (Table. 1). The total amount of these primary factors (known as POA) was highly correlated with eBC
($R^2$= 0.87; Fig. 8a). As a result, the different sources of eBC were evaluated for each factor utilizing a multilinear regression
model, as suggested by Laborde et al., (2013); Zhu et al., (2018) and Poulain et al., (2021), for instance. The following assumes
that the eBC mass is associated with the separate contribution from each OA factor (i.e., e$BC_{HOA}$, e$BC_{BBOA}$, and e$BC_{CCOA}$) at
any time:

$$eBC(t) = eBC_{HOA}(t) + eBC_{BBOA}(t) + eBC_{CCOA}(t) \qquad (3)$$

The eBC emission from each source is expected to be proportionate to the separate source mass concentration generated in
each season ($m_{HOA}$, $m_{BBOA}$, and $m_{CCOA}$, respectively). As a result, the multilinear regression model can be described as follows:

$$eBC(t) = am_{HOA} + bm_{BBOA} + cm_{CCOA} \qquad (4)$$

where a, b, and c are the linear regression coefficients for $m_{HOA}$, $m_{BBOA}$, and $m_{CCOA}$, respectively, that will be applied to evaluate
the contribution of eBC per each POA factor for each season (Table. S3).
CCOA appeared to have the largest source of eBC, contributing half of it (eBC-CCOA 55 %, Table. 1), followed by eBC
associated with BBOA 37 % (eBC-BBOA), while the lowest contribution was found for eBC-HOA (8 %). However, the
contribution of sources to the total eBC strongly depends on the season. Looking at each individual source, the hydrocarbon-
like emissions contributed most to the eBC fraction in the fall (eBC-HOA with 22 %, Table. 1; Fig. 8b), while biomass burning
emissions dominated the eBC in summer and coal combustion emission dominated in winter (eBC-BBOA and eBC-CCOA





with 69 % and 56 %, Table. 1). In the diurnal cycle, contribution to the total eBC of eBC-HOA showed two peaks in the
morning and evening for fall, spring and summer (Fig. S6), reflecting the impact of the traffic rush hours as mentioned in Sect.
3.2.1, and the minimum contributions during the day time due to the effect of lowest emissions and PBL effect (Fig. S3).
However, winter time did not show a strong variation in the diurnal cycle (Fig. S6). This indicates the potential influence of
continuous emissions at the measurement site. Biomass burning combustion with its maximum contribution during the day in
summer (Fig. S6) can be related to a variety of different eBC-POA mass concentrations (Fig. S6b), while the BBOA mass
concentration was almost constant, the other POA mass concentration decreased during the day. Coal combustion showed an
increasing contribution during night time in all the seasons (Fig. S6), especially during the winter time, which further confirms
the enhanced coal combustion emission in winter nights (Fig. S6b).

**3.4 Impact of air mass origin and trajectory analysis**

As mentioned before, the geographical origin of the PM$_1$ chemical species and also PMF components are not only emitted
from the surrounding area but transported. Therefore, to better identify the origin of their sources, trajectory analysis, and their
clustering analysis were applied using the self-developed back-trajectory cluster method (BCLM) (Sun et al., 2020; Ma et al.,
2014; Hussein et al., 2006). A total of fifteen clusters were identified, corresponding to different meteorological conditions
over the course of the year at Melpitz (Fig. 9a). The different clusters can be divided according to the different seasons (CS:
cold season; TS: transition season; and WS: warm season), and meteorological synoptic patterns (ST: stagnant; A1:
anticyclonic with air mass coming from Eastern Europe; A2: anticyclonic with air mass coming from the west; C1: cyclonic
with air mass coming from relatively south; C2: cyclonic with air mass coming from the west and south west). However, the
clustering approach did not consider spring and fall separately, and therefore the transition clusters correspond to both spring
and fall. Regarding this cluster approach, six air masses were identified for the winter season, four air masses for the transition
seasons, and five air masses for the summer season. The number of clusters with their corresponding mean mass concentration
of PM$_1$ chemical species and PMF factors of organics are summarized in Table. 2 and with more details in Tables S3 and S4.

**3.4.1 Winter**

Fig. 9b and 9c illustrate the mass concentration and contribution of PM$_1$ chemical species and PMF factors of organic for each
air mass type at Melpitz based on the type of air masses. For the winter season, the cluster CS-ST corresponds to more
surrounding emission origin with a PM mean value of 21.95 µg/m$^3$, which occurred during 14 % of the total measurement
period. This cluster with the highest mass concentration of LO-OOA to the PM mass (2.73 µg/m$^3$) could confirm the role of
freshly formed SOA originating around the station from primary biomass burning and coal combustion emissions (mass
concentration of 0.97 µg/m$^3$ and 1.89 µg/m$^3$, respectively). Furthermore, nitrate showed a high mass concentration and
contribution in this air mass (5.38 µg/m$^3$ and 25 %, respectively) due to e.g., meteorological conditions and abundant
precursors.



The cluster CS-A1 with the highest mass concentration of PM (29.14 µg/m³) represented Eastern European continental air
masses (passing Poland and the Czech Republic) during anticyclonic flow which occurred during 18 % of the total
measurement period, meaning that Melpitz was under their influence during winter. This air mass, with the highest POA mass
concentration (5.56 µg/m³), especially coal combustion emissions (CCOA and eBC-CCOA with an average mass
concentration of 4.01 µg/m³ and 1.93 µg/m³, respectively), highlight the importance of long-range transported emissions. This
cluster also contained the highest mass concentration of sulphate (5.39 µg/m³) and can support the importance of coal
combustion on sulphate formation, which is known to be strongly emitted by coal power plants (Wierońska-Wiśniewska et al.,

527 2022).

The air mass CS-A2 identified as marine-influenced air with a mean value of 13.39 µg/m³ of PM came from the United
Kingdom with the anticyclonic flow, which occurred during 8 % of the total measurement period. This cluster presented a low
mass concentration of POA and for two OOAs almost the same mass concentration and contribution (Table. S3 and Table.
S4). Since Melpitz is placed away from the coast, therefore the sampling location is affected by aged maritime air masses
(Poulain et al., 2011). Inorganics are dominated by nitrate in this cluster with the high mass concentration (3. 86 µg/m³) and
represent the highest mass fraction (50 % of the total inorganic species).
The CS-C1 air mass with a mean value of 15.99 µg/m³ characteristic of Southern European air mass, came from an industrial
and polluted area starting from Spain and partly crossing Italy with the cyclonic flow, which occurred during 10 % of the total
measurement period. POA mass concentration and contribution were low in this cluster, while SOA, especially MO-OOA,
showed the highest mass concentration of PM over the entire period (3.77 µg/m³) and the highest contribution during the winter
season (24 %). This can be linked to the high sulphate in this air mass (2.99 µg/m³), which showed that the regional influence
by contribution from aged BBOA and CCOA might manifest in MO-OOA (as discussed in Sect. 3.2.2).
Finally, CS-C2a and CS-C2b were both associated with cyclonic and marine influence conditions which only occurred for a
short time (3 % and 2 % of the total measurements, respectively), showing the lowest PM mean value (4.09 µg/m³ and 2.60
µg/m³, respectively). Both of them showed almost the same mass concentration and contribution of POA (Fig. 8a and b; and
Table. S3 and S4). However, similarly to CS-A2, cluster CS-C2a contained a marine component at the beginning point of the
air masses, and in the following time it was dominated by continental areas (France and southern Germany), where due to the
longer time transferring over continent and aging process, it showed more nitrate mass concentration and contribution than
CS-C2b (1.35 µg/m³, 16 µg/m³; and 28 % 14 %, respectively). Whereas CS-C2b started near Iceland with same history of the
air mass over the continent, and in comparison, with CS-C2a, it presented a higher contribution of sulphate (29 % and 19 %,
respectively), which could be associated with aged marine air mass due to the higher contribution of MO-OOA (21 % and 18
%, respectively).
**3.4.2 Transition seasons**
For transition seasons (fall and spring), whereas the four clusters showed a quite similar PM mass concentrations (Fig. 9)
which might be linked to the overall weather situation during these two times of the year, their chemical composition strongly





depended on their origins. TS-A1 and TS-A2 corresponded to two different types of anticyclonic air masses with respective
mean PM mass concentrations of 6.06 µg/m$^3$ and 5.86 µg/m$^3$. Cluster TS-A1 which occurred during 4 % of the total
measurements period, started from Finland, crossing the Estonian, Latvian, Lithuanian and Polish coasts before arriving at
Melpitz. Although it might contain a certain marine component, this cluster mostly followed coastal areas, which means that
in this cluster OA mass concentration dominated PM (2.95 µg/m$^3$). Furthermore, this cluster showed continental and polluted
aspects with the highest LO-OOA mass concentration and contribution during transition seasons (1.03 µg/m$^3$ and 17 %
respectively), which is linked to originating from freshly formed SOA from primary biomass burning and coal combustion
emissions around coastal areas. On the other hand, cluster TS-A2 (4 % of the measurements period) is characterized as a
marine cluster and started from the south of Iceland/Greenland. This cluster showed inorganic as the dominant PM with a high
mass concentration and a mass fraction (3.35 µg/m$^3$ and 58 % respectively). Since Melpitz is influenced by aged marine air
masses, this cluster showed a maximum nitrate mass concentration during the transition seasons (1.54 µg/m$^3$ and a contribution
of 26 %, respectively).
Finally, two other clusters TS-C1 and TS-C2 were two different types of cyclonic air masses in fall and spring time, with mean
PM mass concentrations of 4.69 µg/m$^3$ and 4.94 µg/m$^3$ respectively. These trajectories with different types of marine influenced
air masses occurred for a very short period of time (3 % and 4 % of the total measurements period, respectively). The first one,
TS-C1, started from the Atlantic Ocean near Spain and is associated with a more continental influence, which is why organic
mass concentration and contribution were higher than inorganic. However, The LO-OOA contribution of this cluster was the
highest during this time period (26 %) due to the aging processes of primary organic aerosols especially CCOA, which had a
maximum mass concentration (0.31 µg/m$^3$ and mass fraction of 7 %, respectively). While the second one, TS-C2, was almost
a pure marine cluster, coming from the Norwegian Sea. In opposition to TS-C1, PM was dominated by inorganics in TS-C2,
with a high mass concentration of nitrate (1.35 µg/m$^3$) representing the aging effect due to the long-time transfer over the
continents.

### 3.4.3 Summer

During the summer season, the different clusters showed strong changes in both chemical composition and total mass
concentration. Cluster WS-ST was identified as the local air mass with a mean value of 8.97 µg/m$^3$, which occurred for a short
period, 6 % of the measurement. However, this cluster contained a low POA mass concentration but a maximum contribution
of MO-OOA (32 %), assuming important regional photochemical roles of SOA particles with higher temperatures (Fig. 11)
and enhanced solar radiation (Petit et al., 2015).
Air masses WS-A1 and WS-A2 were two different types of anticyclonic air masses with different directions and different
mean PM mass concentrations. Cluster WS-A1, known as the highest mass concentration during summer time, (16.95 µg/m$^3$
and contribution of 11 % of the measurement period) was the continental air mass which was coming from Eastern Europe
during the anticyclonic flow (starting from Belarus, crossing Poland and the Czech Republic). This air mass included maximum
inorganic and organic especially CCOA mass concentration (1.28 µg/m$^3$) during summer time, which can explain the existing





higher CCOA during summer, and showed the role of long-range transported emissions in the summer season. However, WS-
A2 air mass, with a mean value of 9.48 µg/m³ was a marine-influenced air masse and was coming from the North Sea, which
only occurred for a short period (6 % of the total measurement period).
Moreover, two cyclonic air masses, WS-C1 and WS-C2, were also identified as two different marine clusters. These trajectories
did not occur very often, only 5 % and 3 % of the total measurement period, respectively. The starting point of WS-C1 with a
mean value of 8.41 µg/m³ was the Celtic Sea, but in the following time, it predominantly passed over continental areas (France
and southern Germany), which means it could be aged and the result can be shown in the high mass concentration of nitrate
and sulphate in this cluster (1.63 µg/m³ and 1.86 µg/m³, respectively). Finally, the starting point of WS-C2 with a mean value
of 4.46 µg/m3, was near Iceland, with the lowest PM mass concentration during summer. However, it showed the highest
sulphate contribution (27 %) at this time which could be associated with aged marine air mass like other marine air masses.

### 3.4.4 Cluster seasonality

A parallel comparison can be made between the winter and summer clusters. Clusters CS-A1 and WS-A1 both show the
highest POA contribution dominated by coal combustion, which emphasizes that the origin of this source could be associated
with the transport of the coal power plants emissions from Eastern Europe (e.g. Eastern part of Germany, Poland, Czech
Republic and further countries located in the East). They were not only affected by the winter air quality but also the summer
air quality.
Clusters CS-ST and WS-ST, which were known as local air masses, showed the seasonal effect on the chemical component.
First, the volatility of ammonium nitrate at higher summer temperatures could explain their lower value in summer. Then,
atmospheric photochemical oxidation processes affected the sulphate locally formed emission in summer, which its highest
value over inorganic components during summer can confirm. Not surprisingly, due to the residential heating effect, POA
mass concentration was very high during winter; however, freshly formed SOA originating from biomass and coal emissions
can explain the higher LO-OOA mass concentration in winter.
During the whole period, some marine air masses with cyclonic and anticyclonic flow showed the important roles of aged
marine air masses over the measurement site: a) clusters CS-A2 and WS-A2 with anticyclonic pattern starting from the North
and/or Norwegian Sea, and b) CS-C2a, WS-C1, and TS-C1 starting from the Celtic Sea near Spain, and also CS-C2b and WS-
C2 starting from Iceland, all with cyclonic pattern contain nitrate and sulphate during the transferring over the continental
areas in different seasons.

### 4 Conclusion

Within this study, the change in chemical compositions of non-refractory fine aerosol (NR-PM₁) at the German rural-
background observatory Melpitz was investigated during a one-year period between September 2016 and August 2017, by
applying PMF in a rolling fashion with 14 days window length and a 1-day shift using the SoFi Pro. This method provided the


decomposition of time-dependent factor profiles that were able to better capture the variability of OA sources across seasons,
in particular for LO-OOA. Overall, the averaged total $PM_1$ mass concentration is 10.47 μg/m$^3$ and follows a clear seasonal
pattern, with the highest mass concentration during winter (15.95 μg/m$^3$) and lowest mass concentration during summer time
(6.24 μg/m$^3$). The organic aerosol was the major component accounting for 46 % of total $PM_1$ and showing a strong seasonal
variability ranging from 39 % (in winter) to 58 % (in summer). It was followed by sulphate (15 % and 20 %) and nitrate (24
% and 11 %). The final solution of the PMF rolling approach for OA source apportionment enabled the identification of five
factors throughout the one-year measurements of OA; HOA, BBOA, CCOA, LO-OOA, and MO-OOA.

Generally, in Melpitz, HOA as a minor source of OA (6 % of the contribution of total organic mass) was associated with: a)
low traffic emissions, b) household heating in winter, and c) the central heating for hot water production for all the seasons
which showed a small increasing mass concentration pattern toward cold months (winter and summer: 0.36-0.23 μg/m$^3$). The
HOA night time mass concentration was not affected by the seasons, which indicates the presence of a continuous emission
source. Biomass burning emissions (BBOA) representing 7.9 % of the contribution of total organic mass showed a seasonal
effect, emphasizing the impact of house heating during winter (winter and summer: 23 % and 8.7 %). This highest mass
concentration during the winter time showed the descending pattern from night time to day time due to domestic heating
activities and the planetary boundary layer effect; however similar to HOA, the presence of BBOA during summer was due to
central heating which uses multiple fuel types in the Melpitz area. The most dominant anthropogenic source was associated
with coal combustion (CCOA) with a 15.4 % contribution of total organic mass and 55 % of eBC with the highest mass
concentration and contribution of PM during winter rather than summer (1.58-0.30 μg/m$^3$). Although a certain fraction of
CCOA could be linked to surrounding domestic heating (van Pinxteren et al., 2023), it is rather associated with power plant
emissions and long-range transport all year round. Using the correlation between HOA, BBOA, and CCOA with eBC, a
multilinear regression approach was applied to perform the source apportionment of eBC. This analysis highlighted eBC
contribution related to the source of HOA (8 % of the total eBC), BBOA (37 % of the total eBC), and CCOA (55 % of the
total eBC), which showed the CCOA as the largest source of eBC during the measurement period. Moreover, from the seasonal
source apportionment, CCOA presented the largest fraction (56 % of the total eBC) during winter, while the highest fraction
is attributed to BBOA for summer time (69% of the total eBC). LO-OOA and MO-OOA referred to oxidized oxygenated
organic aerosol (32.4 % and 38.4 % of the contribution of total organic mass, respectively), were identified as a secondary
organic aerosol with the highest mass concentration during the cold months (fall: 2.13 μg/m$^3$ and winter: 2.25 μg/m$^3$,
respectively) and the lowest mass concentration during the warm months (spring: 1.24 μg/m$^3$ and summer: 1.44 μg/m$^3$,
respectively). LO-OOA mass concentration decreased during the day due to dilution, and the evaporation process resulted in
aging into MO-OOA.

A combination of pollution wind rose and cluster analysis was used to better understand the origin of the aerosol reaching the
station. Overall, Melpitz is influenced by fifteen types of air masses, such as long-range continental, marine, and surrounding



emissions. During winter and summer time, easterly continental air masses, CS-A1 and WS-A1 with an anticyclonic pattern
come from Eastern Europe and showed a significant particle mass concentration, especially high POA (and CCOA) mass
concentration at the measurement site. Marine clusters, mostly coming from the south/west/north side with aged marine air
masses including nitrate and sulphate, also have important roles in the PM mass concentration at the Melpitz site over the
entire period (winter: CS-A2, CS-C2b, and CS-C2a, transition: TS-C, TS-A2 and TS-C2, and summer: WS-Ca, WS-C2, and
WS-A2). However, the surrounding emissions are recognized as another important source of emissions which include high
organic and inorganic components during winter and summer (CS-ST and WS-ST, respectively).

Our results emphasize the importance of the long-range transported emissions of coal combustion related aerosol particles
regardless of the season, which supports that the main CCOA source is related to coal power plants emissions. However, coal
power plants emissions not only affect the surrounding air quality but can also be transported over long distances. It is important
to note that the overall coal combustion mass concentration presented here can certainly be underestimated since the identified
CCOA factor is associated with freshly emitted organic aerosol and no factor associated with potential aged coal combustion
was identified. Because coal still is an important energy source in the European energy mix (68.4 % of all energy in the EU
was produced from coal, crude oil, and natural gas, Energy Statistics - an Overview - Statistics Explained, 2022) as well as on
a global scale and also that it still will be in used for the coming decades (until 2040, Europe's Coal Exit - Europe Beyond
Coal : Europe Beyond Coal, 2022), further research should be done on the identification of coal emissions across Europe in
order to better understand its atmospheric aging processes.

**Acknowledgements**
This work is supported by the COST action CA16109 Chemical On-Line cOmpoSition and Source Apportionment of fine
aerosoLs (COLOSSAL), the SNF COST project SAMSAM IZCOZO_177063., by the infrastructure projects ACTRIS (EU
FP7, grant 262254), the RI-URBANS project under grant NO. 101036245, the ERA-PLANET, and transnational projects
SMURBS and iCUPE (grant agreement NO. 689443), and ACTRIS-2 (Grant 654109).

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





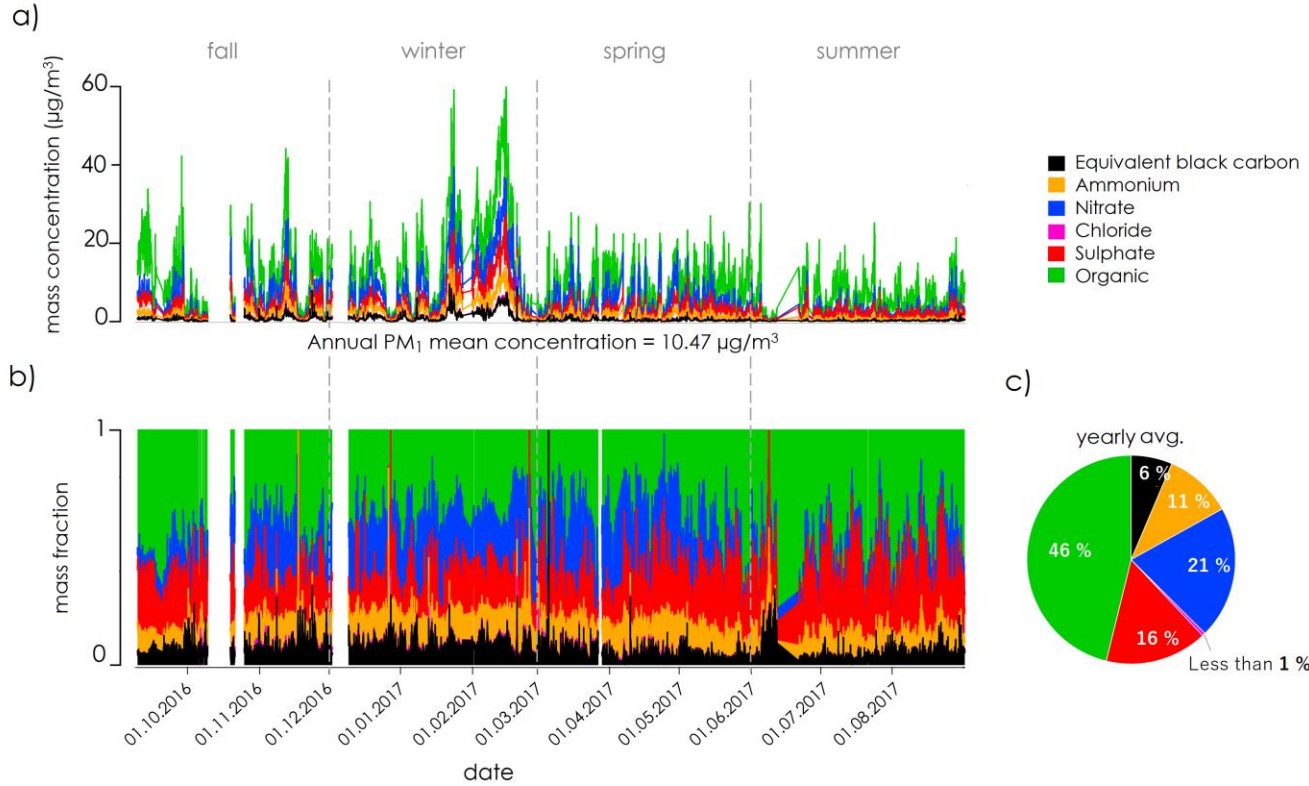

**Fig. 1: Time series of a) the particulate PM$_1$ chemical composition, b) the corresponding mass fraction and c) average contribution of each chemical component (Time is in UTC).**





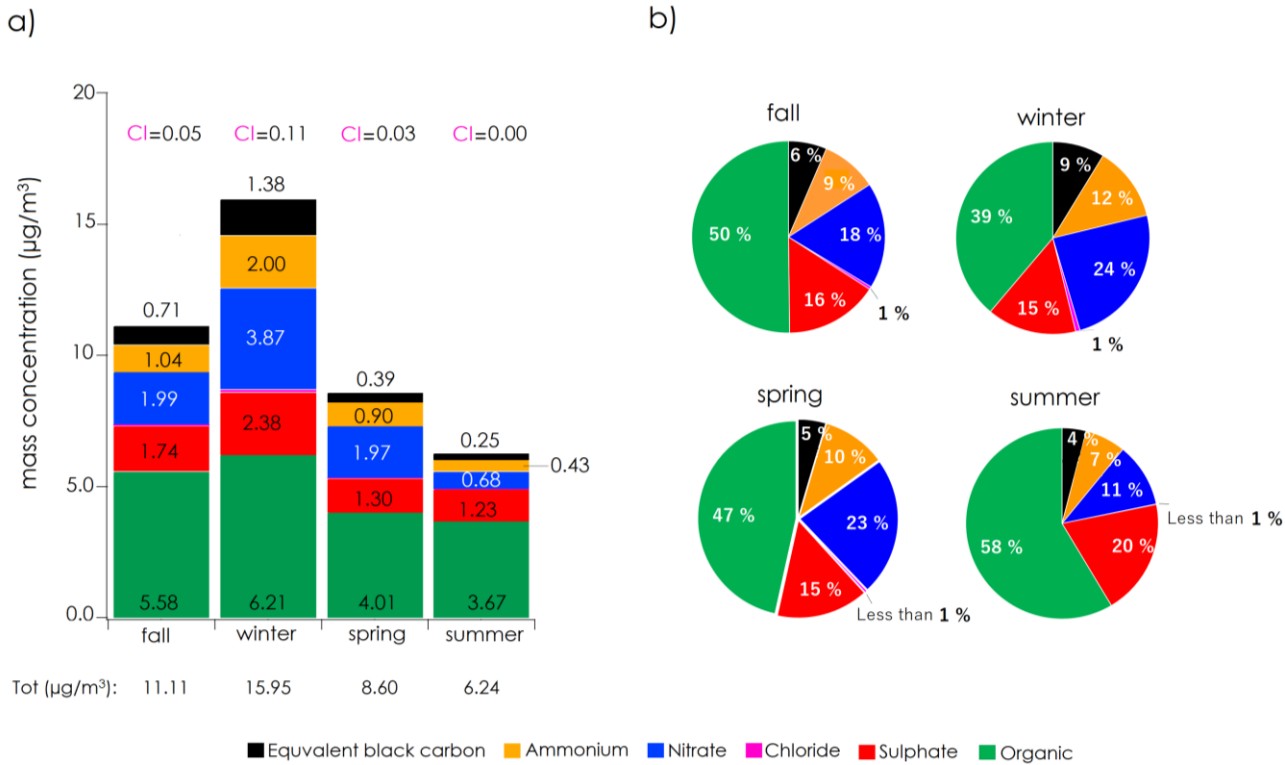

**Fig. 2: seasonal variation of PM$_1$ a) absolute mass concentration and b) mass fraction.**





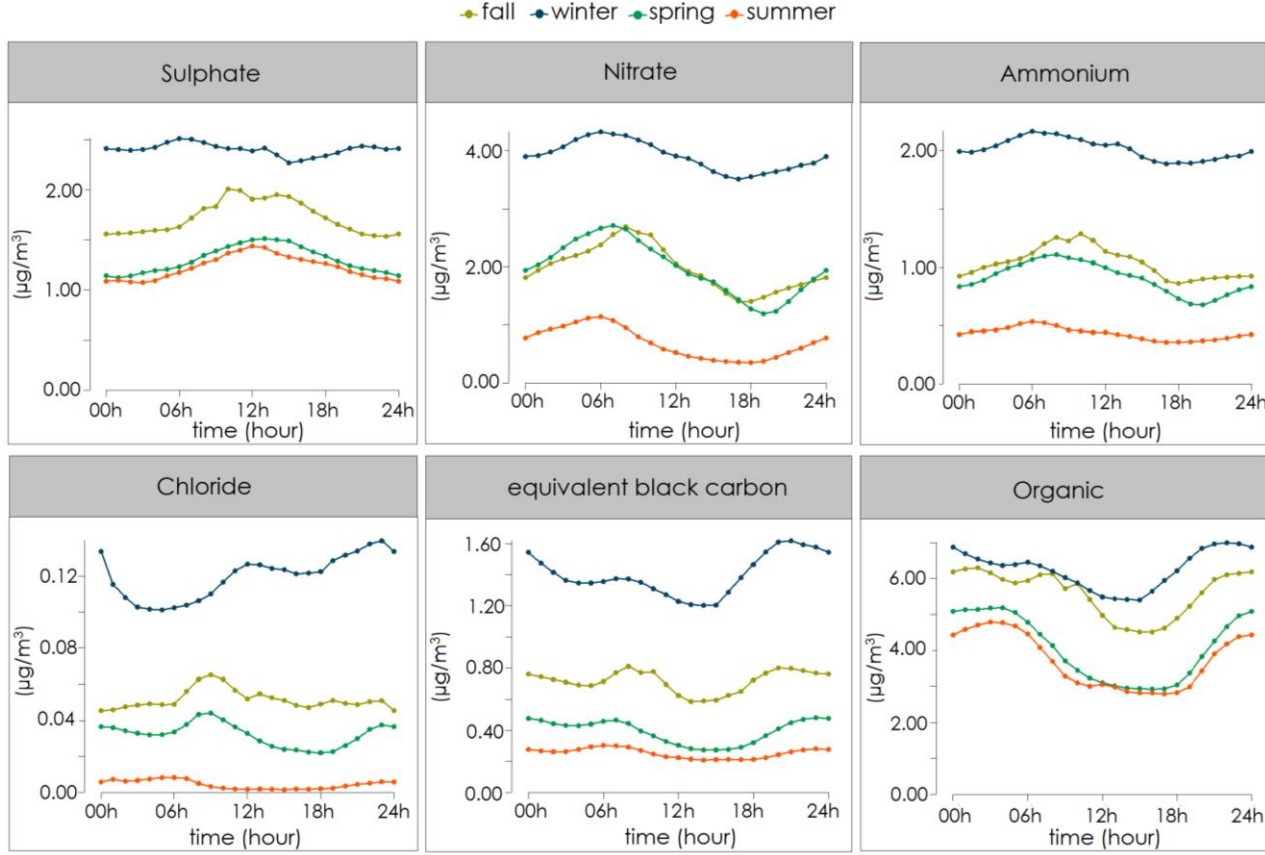

**Fig. 3: Seasonal diurnal cycle of $PM_1$ for ACSM organic and inorganic species (Time is in UTC).**

**Table. 1: Seasonal/yearly mass concentration of each ACSM species, each PMF factors, contribution of the different POA-PMF-eBC, and correlation of each factors with related species; $PM_1$.**

| | Species/ Factors | Fall | Winter | Spring | Summer | Yearly |
|---|---|---|---|---|---|---|
| **ACSM** **($\mu g/m^3$)** | **Org** | 5.58 | 6.21 | 4.01 | 3.67 | 4.84 |
| | $SO_4^{2-}$ | 1.74 | 2.38 | 1.30 | 1.23 | 1.67 |
| | $NO_3^-$ | 1.99 | 3.87 | 1.97 | 0.68 | 2.16 |
| | $NH_4^+$ | 1.04 | 2.00 | 0.90 | 0.43 | 1.11 |
| | $Cl^-$ | 0.05 | 0.11 | 0.03 | 0.00 | 0.05 |
| **MAAP** **($\mu g/m^3$)** | **eBC** | 0.71 | 1.38 | 0.39 | 0.25 | 0.66 |
| **PMF** **($\mu g/m^3$)** | **HOA** | 0.35 | 0.36 | 0.27 | 0.23 | 0.30 |
| | **BBOA** | 0.36 | 0.72 | 0.27 | 0.21 | 0.39 |
| | **CCOA** | 0.72 | 1.58 | 0.47 | 0.30 | 0.77 |
| | **LO-OOA** | 2.13 | 1.95 | 1.24 | 1.26 | 1.62 |
| | **MO-OOA** | 2.21 | 2.25 | 1.82 | 1.44 | 1.92 |
| **eBC** **($\mu g/m^3$)** | **eBC-HOA** | 0.16 | 0.19 | 0.03 | 0.04 | 0.05 |
| | **eBC-BBOA** | 0.34 | 0.38 | 0.17 | 0.15 | 0.25 |
| | **eBC-CCOA** | 0.23 | 0.74 | 0.16 | 0.02 | 0.37 |
| **eBC** **(%)** | **eBC-HOA** | 22 | 15 | 9 | 18 | 8 |
| | **eBC-BBOA** | 47 | 29 | 47 | 69 | 37 |
| | **eBC-CCOA** | 31 | 56 | 44 | 13 | 55 |





| | | | | | | |
|---|---|---|---|---|---|---|
| | HOA/eBC | 0.49 | 0.52 | 0.34 | 0.24 | 0.33 |
| | HOA/NO$_x$ | 0.23 | 0.12 | 0.32 | 0.23 | 0.17 |
| | BBOA/Levo. | 0.19 | 0.59 | 0.09 | 0.07 | 0.54 |
| Correlation | BBOA/eBC | 0.62 | 0.81 | 0.48 | 0.42 | 0.77 |
| (R$^2$) | CCOA/eBC | 0.65 | 0.85 | 0.49 | 0.30 | 0.82 |
| | CCOA/Cl$^-$ | 0.40 | 0.41 | 0.18 | 0.15 | 0.46 |
| | LO-OOA/ NO$_3^-$ | 0.22 | 0.59 | 0.01 | 0.12 | 0.19 |
| | LO-OOA/ SO$_4^2$ | 0.36 | 0.55 | 0.00 | 0.02 | 0.23 |
| | MO-OOA / SO$_4^2$ | 0.58 | 0.47 | 0.34 | 0.42 | 0.44 |
| | MO-OOA/ NO$_3^-$ | 0.24 | 0.47 | 0.16 | 0.24 | 0.31 |

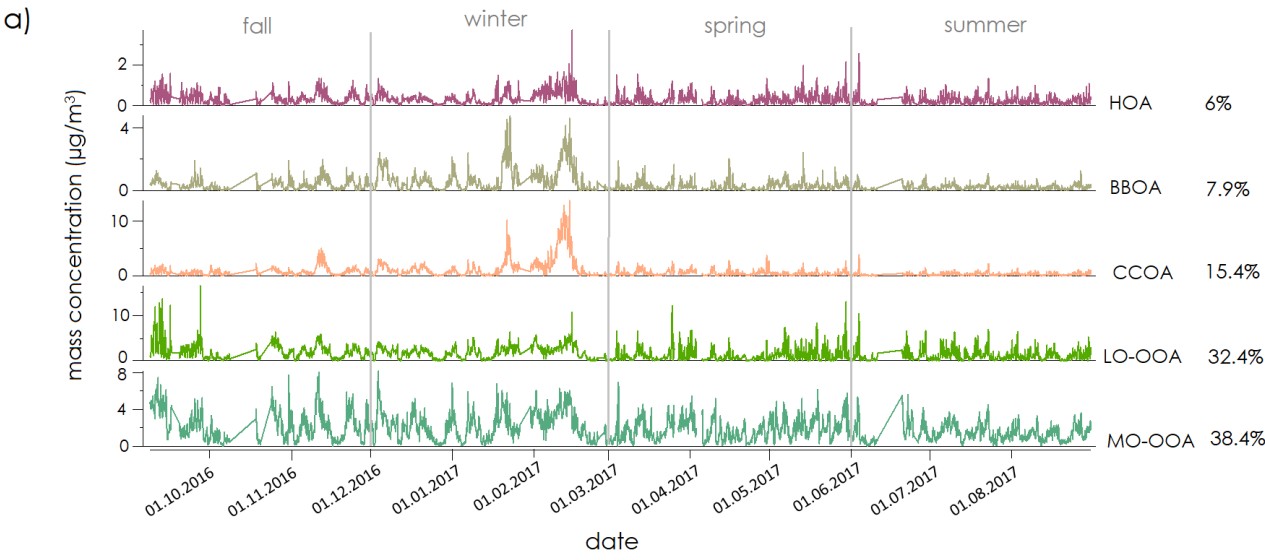

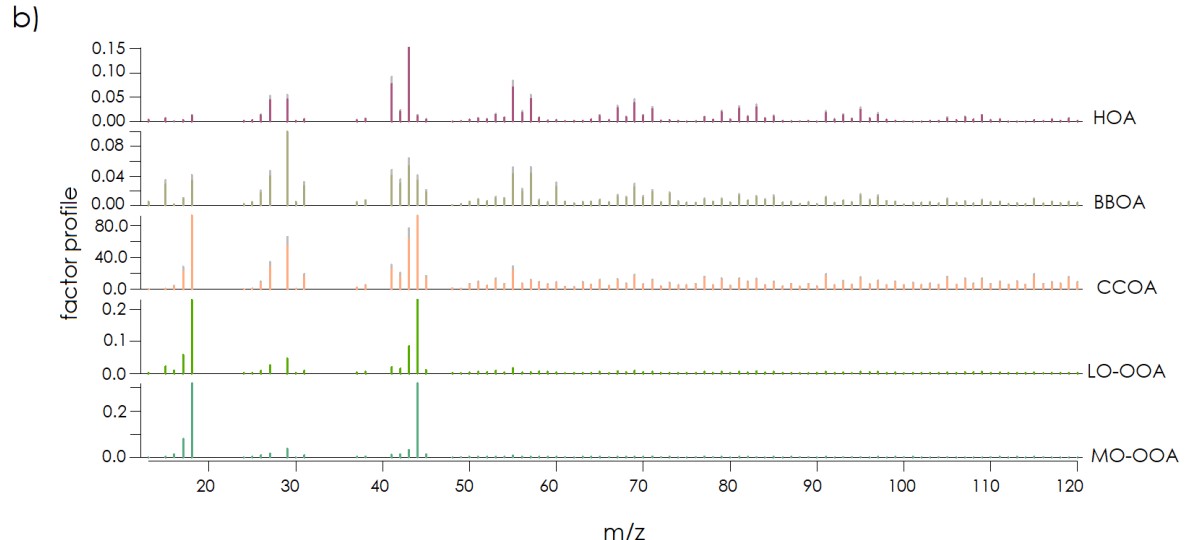

**Fig. 4: Overview of averaged PMF (ME-2) results, a) time series, and b) mass spectral profile of organic PMF factors. (Time is in**

**UTC)**







**Fig. 5: Seasonal variation of a) mass concentration, b) mass fraction of PMF source factors.**





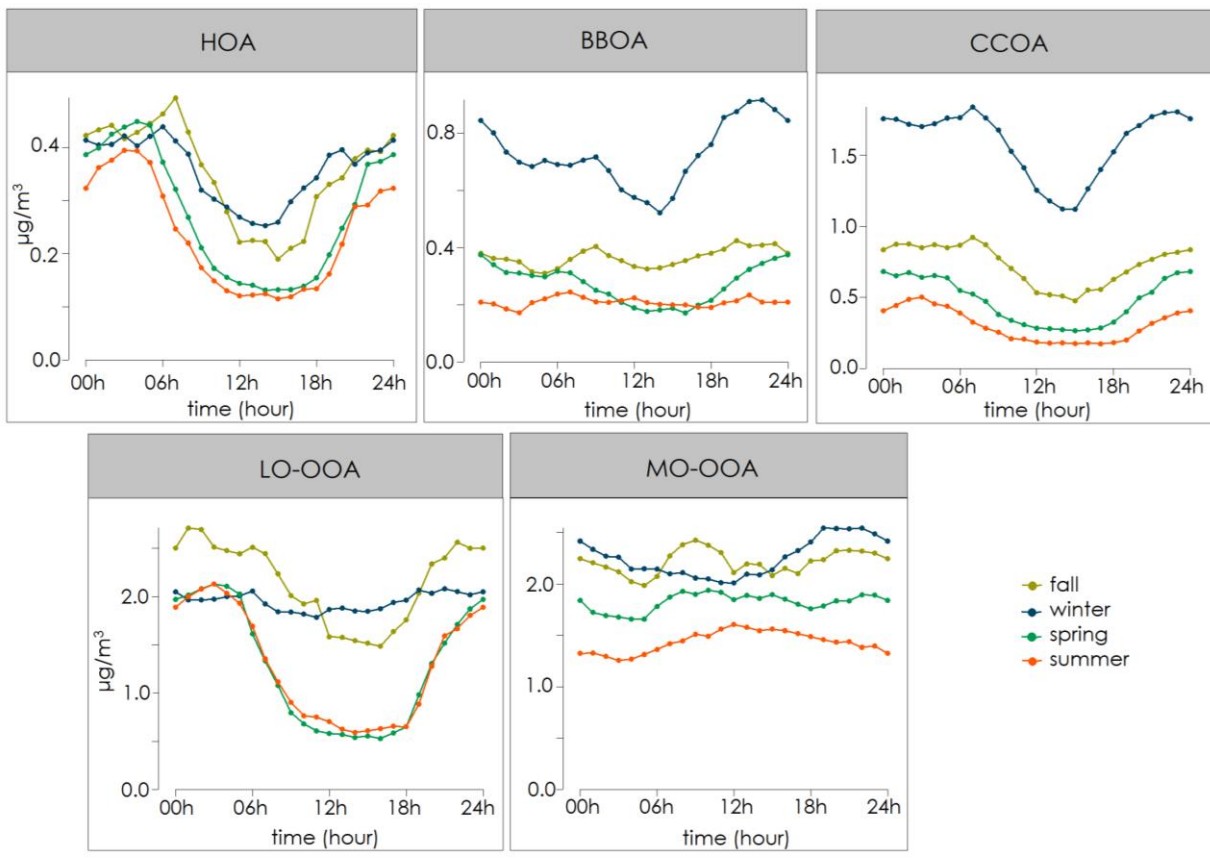

**Fig. 6: Seasonal diurnal cycle (hourly averages) of the organic components HOA, BBOA, CCOA, LO-OOA and MO-OOA in UTC time.**





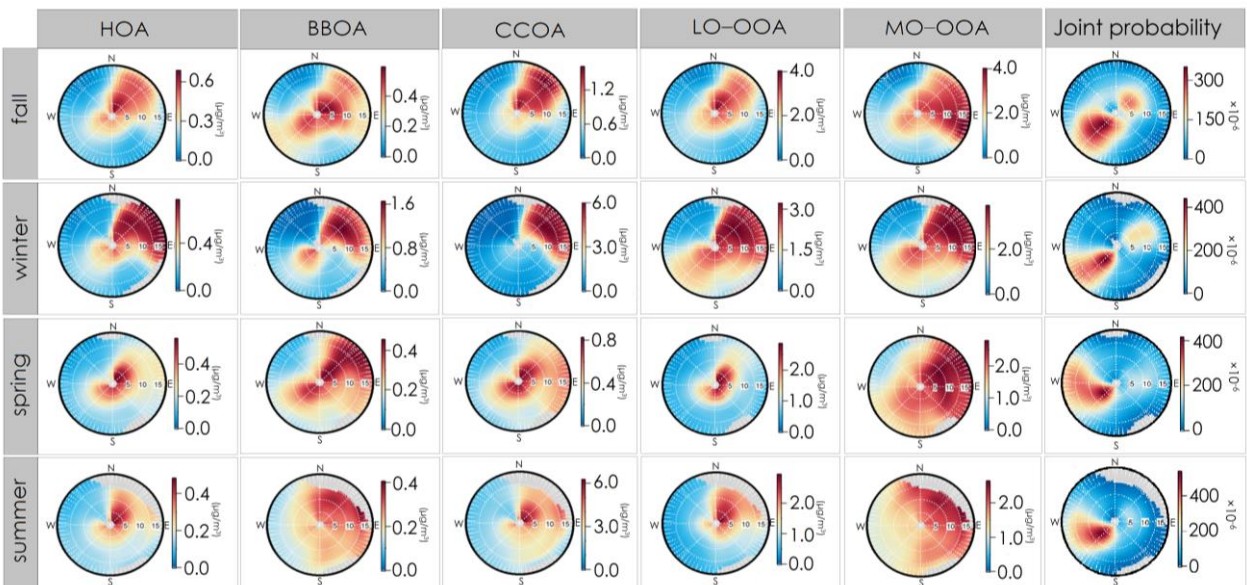

**Fig. 7: Seasonal wind roses and NWR plots for the different PMF factors (in µg/m³).**

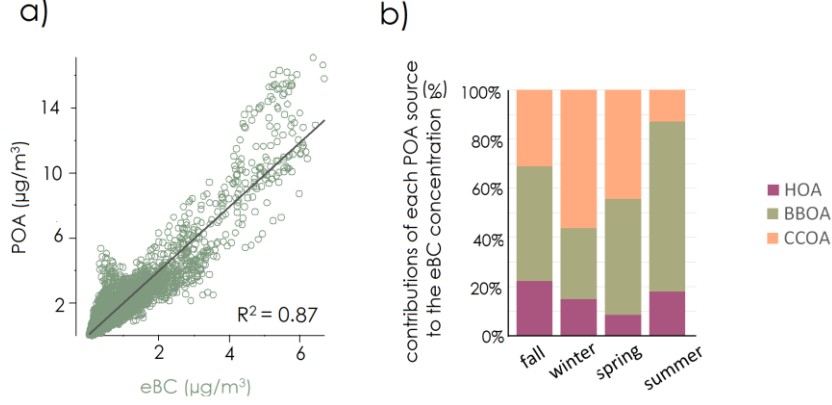

**Fig. 8: Contribution of the three POA factors to the mass concentration of eBC, a) scatter plots POA vs eBC and b) contributions of sources to the eBC mass concentration.**





**Fig. 9: a) air mass classification based on 13-years backward trajectories cluster analysis at 12:00 UTC, b) influence of air mass to the PM$_1$ data and PMF factors, and c) contribution of them which averaged from 10:00 to 14:00 UTC.**



**Table. 2: Main statistical details of the 15 air mass types for total PM₁**

**(CS=Cold Season, WS=Warm Season, ST=Stagnant, A=Anticyclonic, C=Cyclonic).**

| Main season | Airmass type | Wind direction | Vorticity | Frequency (%) | Total mean (µg/m³) |
|---|---|---|---|---|---|
| Winter | CS-ST | Stagnating | Anticyclonic | 14 | 21.95 |
|  | CS-A1 | East | Anticyclonic | 18 | 29.14 |
|  | CS-A2 | West | Anticyclonic | 8 | 13.39 |
|  | CS-C1 | South | Cyclonic | 10 | 15.99 |
|  | CS-C2a | South West | Cyclonic | 3 | 04.09 |
|  | CS-C2b | West | Cyclonic | 2 | 02.60 |
| Transition (Spring/ Fall) | TS-A1 | North East | Anticyclonic | 4 | 06.06 |
|  | TS-A2 | West | Anticyclonic | 4 | 05.86 |
|  | TS-C1 | South West | Cyclonic | 3 | 04.69 |
|  | TS-C2 | North West | Cyclonic | 4 | 04.94 |
| Summer | WS-ST | Stagnating | Anticyclonic | 6 | 08.97 |
|  | WS-A1 | South East | Anticyclonic | 11 | 16.95 |
|  | WS-A2 | North West | Anticyclonic | 6 | 09.48 |
|  | WS-C1 | West | Cyclonic | 5 | 08.41 |
|  | WS-C2 | West | Cyclonic | 3 | 04.46 |

1120