# Peer review of "A one-year ACSM source analysis of organic aerosol particle contributions from anthropogenic sources after long-range transport at the TROPOS research station Melpitz"

_Atmospheric Chemistry and Physics, 2022_

## Author Comment (AC1)

**Responses to the reviewer's comments on**
**"A one-year ACSM source analysis of organic aerosol particle contributions from anthropogenic sources after long-range transport at the TROPOS research station Melpitz" by Atabakhsh et al.**

Please find below the response to reviewer #1:
* * *
**Reviewer #1 (R1)**:

We thank the reviewer for his insightful review and constructive comments. We highly appreciate your time in reviewing the manuscript. All the comments/suggestions were taken into consideration and incorporated in the revised manuscript, which has improved the quality of the revised manuscript. The point by point response to all the comments and suggestions of reviewer #1 (R1) is provided in the following sections. For clarity, the reviewer's comments are provided in blue (RC), the author's comment (AC) is in black, and the revised parts of the manuscript are shown in red.

Atabakhsh et al. have reported a nice study of one-year ACSM/MAAP measurement. Using rolling PMF and a multilinear regression model, they have conducted detailed source appointments for non-refractory aerosol components and eBC. In addition, to better identify the origin of aerosol sources, clustering analysis was also applied using the back-trajectory cluster method. Based on these approaches, the variations of chemical composition, aerosol mass concentrations, and diurnal cycles among the meteorological seasons were carefully examined and discussed, especially the comparison was performed between the cold and warm seasons. I have a couple of comments on the determination of the emission sources of sulfate, nitrate and org. The comments are not serious but need to be clarified and justified. Overall, the manuscript is well written and easy to read. I would recommend its publication after my comments are addressed.

**AR:** Thank you for your generally positive view on our submission. We have revised in our manuscript, see below at the individual points. We think this way the discussion of the results as well as their interpretation was much improved.

**General comments:**

**R1C1:** The interpretation of sources and diurnal cycles of aerosol components relies heavily on the wind rose patterns and NWR analysis in Sections 3.1-3.3. If I understand correctly, both techniques can only determine local emission sources. However, you have claimed the significance of long-range transported aerosol sources for different measured aerosol species in this study. The statement appears bit abrupt and needs more clarification. Since you have conducted nice back-trajectory and cluster analysis later, I would suggest you already cite those results to support your conclusions in Sections 3.1-3.3.

**AC1:** We thank the reviewer for the comment. The non-parametric wind regressions model (NWR) can be used to investigate not only the local but also the transported emission sources as it was already done in for example (Marin, et al., 2019). With this regard, high chemical species mass concentrations observed at low wind speeds, can be recognized as local sources, whereas high mass concentrations associated to higher wind speeds correspond to more transported ones coming from this specific wind sectors. Consequently, the NWR analysis not only provides information on local emissions but also gives some details on the transported sources associated with a main prevailing wind at high wind speed. Nevertheless, we have to recognize that this approach did not provide any information on the distance as well as the location of the emissions' areas. In order to clarify the statement, the following lines have been added to the NWR definition:

- **Line 214-216 (revised version):** Non-parametric wind regressions (NWR) were used to approximate the OA source concentrations at a given wind direction and speed (Henry et al., 2009) in order to investigate not only the local but also the prevalent wind sector associated with transported emission sources (Marin, et al., 2019).

Since the wind direction did not provide any information on the air mass origin, cluster analysis on air mass back-trajectories was used to identify the long-range transported emissions and it was discussed in Sect. 3.4. However, we briefly included them in Sect. 3.1-3.3, we decided to keep Sect. 3.4 (Seasonal air mass clustering) as an overview of this study which comprehensively discussed the origins of chemical species and PMF sources. Therefore, some parts of Sect. 3.1-3.3 have been revised as follows:

[revised manuscript text omitted]

**Specific comments:**

**R1C2:** L16, How about nucleation? Should it be a source of secondary aerosol?

**AC2**: We thank the reviewer for the comment. This is true that one of the mechanisms for the formation of SOA could be organic nucleation (Zhu and Penner, 2019), since the condensation of organic species, plays an important role in the growth of newly formed particles into larger ones. Moreover, small particles do not play a significant role in the total particle mass. Only when the newly formed particles reach a significant diameter (approx. 100 nm), they can significantly influence the overall mass. At this point, the competition between the contribution of condensing organic species to the newly formed particle growth and the pre-existing particles is hard to distinguish. Therefore, we preferred to use the SOA, which indirectly includes all the different processes of SOA formation.

**R1C3:** L48-49, $PM_1$ also has an effect on air quality, right?

**AC3:** $PM_1$, as one of the air pollutants itself, it is not necessary to address it actually affects the air quality. To clarify the reason why $PM_1$ is important, the text has been modified as follows:

**Line 45-49 (revised version):** The submicronic particles known as $PM_1$ (particles with an aerodynamic diameter less than 1 µm), not only have a negative impact on human health (Pop and Dockery, 2016; Daellenbach et al., 2020) but also have a significant effect on visibility (Shi et al., 2014) and climate (Shrivastava et al., 2017). It is ability to penetrate to respiratory system make it more dangerous, therefore more relevant to mitigate adverse health impact.

**R1C4:** Sec. 2.2 How did the temperature vary at the measurement station where ACSM was located, since the ambient temperature may affect the instrument sensitivity from season to season?

**AC4:** Since all instruments were situated in a container with air conditioning, the temperature of the instruments have been exposed to fairly constant over seasons. Therefore, we consider the influences by ambient temperature to the instrument sensitivities are negligible in our case.

**R1C5:** Sec.3.1 Since ACSM is associated with a couple of measurement uncertainties, such as the determinations of collection efficiency and response factor, it would make a lot of sense to compare ACSM derived concentration to other instruments in the parallel measurements.

**AC5:** The intercomparing of the ACSM results with collocated instruments including SMPS, off-line $PM_1$, and $PM_{2.5}$ for the total particulate mass, water-soluble ions and OC/EC was already discussed in deep detail in our previous work (Poulain et al., 2020), it appears to be out of the scope of this paper to repeat this work and it was only referred to it on the preprint manuscript in line 117-118 preprint version ("Details on the QA/QC for this dataset can be found in Poulain, et al., (2020)"). In order to better refer to our previous work, the sentence line was changed as follows:

**Line 122-125 (revised version):** The quality assurance of the ACSM measurements was performed by comparing them with collocated measurements including MPSS, and high-volume filter samples ($PM_1$ and $PM_{2.5}$) for the total particle mass concentration, water-soluble ions (nitrate, sulphate, and ammonium), as well as OC/EC. Details on the QA/QC and instrumental uncertainties can be found in Poulain et al., (2020).

**R1C6:** L228-231, As you have mentioned in the nitrate results in the next paragraph, nitrate concentration dropped in the afternoon due to the vertical mixing. Similar phenomenon is expected for the sulfate species too. However, a flat diurnal cycle was observed in winter when photochemistry was weak. Did it indicate that, except for long-range transport, there were additional sources for sulfate around noon in wintertime? Can you comment on it? In addition, you observed a high peak at noon in other three seasons, and you interpreted it to be the photochemistry of $SO_2$. I just wonder if enough OH are available for $SO_2$ photochemistry to explain such a diurnal peak given the effect of boundary layer mixing? A rough estimation on this point would not go amiss.

**AC6:** It is true that vertical mixing can influence the ambient mass concentration, leading to a decrease of the mass concentration by a "dilution effect". However, it is not the only process that can influence the ambient concentration of chemical species. For example, it is well known that ammonium nitrate is a semi-volatile compound in equilibrium between the particle phase ($NH_4NO_3$) and the gas phase ($NH_3$ and $HNO_3$), which is affected by the gas phase concentration of $NH_3$ and $HNO_3$, temperature, and hygroscopicity. Therefore, the decrease of the nitrate mass concentration during day time is not only influenced by the vertical mixing but can also be affected by its gas-to-particle equilibrium. Thermodynamics of the ammonium nitrate at Melpitz was already discussed in Poulain et al., (2011).

On the opposite, sulfuric acid is a less volatile species than $HNO_3$ and will condense on the particle phase, leading to ammonium sulphate, which will be mostly affected by the dilution effect due to the change of the mixing layer height. Consequently, the mentioned drop of the ammonium nitrate in the afternoon can be associated with a change in the ammonium nitrate partitioning, while ammonium sulphate stays in the particle phase and is constant during the winter. Moreover, the higher sulphur dioxide ($SO_2$) concentration in winter, can also lead to photochemically formed $H_2SO_4$ and finally in particulate sulphate.

Furthermore, Größ et al., (2018) presented the relationship between $SO_2$, the hydroxyl radical, OH, and sulfuric acid ($H_2SO_4$) in the boundary layer for different cases at the Melpitz station (June 2010, May 2008, June 2010, and August 2008). They highlighted that high sulfuric acid concentrations were caused primarily by the high level of $SO_2$ and hydroxyl radical •OH with significant solar radiation around noon. This could simply explain the high peak at noon not only in the winter time but also the in the other season as well.

**R1C7:** L231-232, You should make it clear that it is valid only for the same season that a high sulfate concentration was shown at low wind speed, but does not hold when you compare it among different seasons.

**AC7:** This comment was already answered in our answer to the general comment of the reviewer. The mentioned text has been revised as follows:

**Line 286-291 (revised version):** Furthermore, the NWR plots (Fig. S3) show that during the winter time, sulphate mostly comes from the north and east sectors with wind speeds above 5 m/s which can be associated with dominant transported sulphate sources. Although the eastern wind sector remains visible for the sulphate in the summer time, the high concentrations of sulphate can be observed during periods with low wind speed and without a specific wind sector; which corresponds to local sulphate formation. Sect. 3.4 will go into detail about the long-range transported emissions later on.

**R1C8:** L248-250, You had very good discussion on the local contributions to eBC and org in section 3.1.2, did the local source contribute to nitrate during winter season? As far as I understand, NWR only determines local sources, nor the long-range transported ones (Fig. S2). If you argue the dominant sources of nitrate from long-range transport, you might need to justify it. Furthermore, you cluster analysis shows a high local nitrate mass concentration (CS-ST), should this suggest a local source of nitrate?

**AC8:** We thank the reviewer for the comment. The discussed lines are: "In winter, ammonium-nitrate remains mainly in the particle phase (Seinfeld and Pandis, 2006) and, like sulphate, arrived at the measurement site due to the long-range transported emissions which not only came from the north-eastern but also south-western flow, describing higher mass concentrations for nitrate and ammonium (Fig. S2).", pre-print version, which we are going to answer to the comment in two parts:

- Since the statement is talking about ammonium nitrate during winter time, the word 'mainly' just emphasizes that ammonium nitrate mostly exists in particle phase at low temperature (Spindler et al., 2010) due to its volatility, and we did not claim about the major source of nitrate here. However, in the lines 307-312 (revised version), the long-range transported emissions of nitrate have been mentioned, since the study of Spindler et al. (2010) confirms the arriving long-range transported particles (including sulphate, nitrate, ammonium, and carbonaceous materials) from the other side toward Melpitz site.

- As we already discussed in response to 'General comments', the NWR model works with local wind measurements but can determine the local and transported emissions. However, besides the local source for nitrate, it shows two different directions with a higher wind speed during winter time rather than other seasons (Fig. S8). This high wind speed can bring the particles from pollution sources toward the measuring site. Furthermore, to confirm the long-range transported emissions, cluster and back-trajectory analysis are used which is widely explained in Sect. 3.4. Nevertheless, in the winter time, all results from NWR, cluster, and back-trajectory analysis present two sources:

    a) local source as a result of both the NWR plot (Fig. S3) and cluster analysis with CS-ST air mass (Fig. 9).

[Figure]

**Fig. S3: Seasonal NWR plots for the different chemical compositions (in µg/m³). PM₁ is the average of all the compositions.**

[Figure]

**Fig. 9: a) air mass classification based on one-year backward trajectories cluster analysis at 12:00 UTC, b) influence of air mass to the PM₁ data and PMF factors, and c) contribution of them which averaged from 10:00 to 14:00 UTC.**

b) long-range transported emission which not only can be seen from the NWR plot (Fig. S3) with higher wind speed in two different directions but also cluster CS-A1 as eastern air mass and CS-A2 north-western air mass (Fig. 9) confirm a high mass concentration during winter time, representing the aging effect due to the long-time transfer over the continents.

To avoid misunderstanding, the statement has been revised as following:

**Line 307-315 (revised version):** Nitrate profiles from NWR plots (Fig. S3) present two different wind directions for the whole period which might be associated with transported nitrate from Leipzig and Torgau (50 km in the south-west and 7 km in the north-east of Melpitz, respectively) with higher wind speed. Since the reaction pathway of OH and $NO_2$ can result in nitrate formation (Yang et al., 2022), this mechanism can be linked to traffic emissions in residential areas. These long-range transported sources together with locally formed emissions could describe higher mass concentrations for nitrate and ammonium due to e.g., meteorological conditions and abundant precursors in winter time. However, in winter, ammonium-nitrate remains mainly in the particle phase (Seinfeld and Pandis, 2006) since it can totally be changed from gas to particle phase at lower temperature (Spindler et al., 2010). High values of nitrate and ammonium in spring time are linked to agronomical fertilization (Stieger et al., 2018).

**R1C9:** L254-256, Based on my rough estimation of chemical composition during winter (Fig. 2), nitrate and sulfate were fully neutralized by ammonium as ammonium nitrate and sulfate. It sounds like the contribution of organic nitrate was negligible in this study.

**AC9:** Thank you for the comment. We agree with the reviewer about the neutralization process, since the previous work (Poulain et al., 2020) estimated the neutralization state of the particles assuming a full neutralization by nitrate and sulphate, and confirmed that particles can be considered as fully neutralized during the measurement for Melpitz site, which also agrees with previous AMS measurements made at Melpitz (Poulain et al., 2011). Furthermore, Kiendler- Scharr et al. (2016) have already shown that organo-nitrate can play a significant role in the nitrate signal. However, it was possible to quantify them only within high resolution AMS. Since Q-ACSM is working at a unit mass resolution (UMR), it is not possible to distinguish nitrate from organic signals at $m/z$ 30 ($CH_2O^+$ and/or $C_2H_6^+$) and $m/z$ 46 ($CH_2O_2^+$, $C_2H_6O^+$) ratios. Therefore, estimating the organo-nitrate would only introduce uncertainties to measurements, therefore, we did not consider to conduct this analysis in this study.

To make the statement clearer, we added the following lines to the ACSM section of manuscripts:

**Line 126-132 (revised version):** The ACSM ammonium mass concentration mainly corresponds to ammonium nitrate and ammonium sulphate salts. Previously by Poulain et al. (2020), the neutralization state of the particles was estimated for datasets assuming complete neutralization by nitrate, sulphate, and chloride. Therefore, the particles are neutralized when considering nitrate, ammonium, and sulphate in this study. Furthermore, the significant role of organo-nitrate and organo-sulphate on signals of nitrate and sulphate is not negligible (Kiendler- Scharr et al. (2016). Since the Q-ACSM is working at a unit mass resolution (UMR), it is not possible to distinguish nitrate and sulphate from organic. Therefore, estimating the organo-nitrate would only introduce uncertainties to measurements, therefore, we did not consider to conduct this analysis in this study.

**R1C10:** L281-283, Traffic is a good source of nitrate, how does it affect your observed nitrate in this study?

**AC10:** Thank you for the comment. The statement in these lines (317-320 preprint version), mainly explains the two peaks of eBC-$PM_1$ in the morning and evening based on possible traffic emissions from federal street (B 87) which is located approximately 1.5 km away from the station in the north direction. This statement concluded mostly based on the similarity of diurnal profiles of eBC-$PM_1$ and nitrogen oxides which is known as a tracer for traffic emissions (Fig. S4) and not referred to as particulate nitrate.

[Figure]

**Fig. S4: Seasonal diurnal cycle of Temperature, sun radiation, Sulphur dioxide, Nitrogen Oxides, and Ozone.**

Regarding nitrate source, nitrate formation from the reaction of OH and $NO_2$ pathway is well-known (e.g. Yang et al., 2022). The observed nitrate in our study (Fig. S3) is related to two main wind directions for all the seasons, which can be associated to transport from Leipzig and Torgau (50 km and 7 km distance from Melpitz, respectively). Therefore, we added the following statement to the nitrate discussion in Sect. 3.1:

**Line 307-312 (revised version):** Nitrate profiles from NWR plots (Fig. S3) present two different wind directions for the whole period which might be associated with transported nitrate from Leipzig and Torgau (50 km in the south-west and 7 km in the north-east of Melpitz, respectively) with higher wind speed. Since the reaction pathway of OH and $NO_2$ can result in nitrate formation (Yang et al., 2022), this mechanism can be linked to traffic emissions in residential areas. These long-range transported sources together with locally formed emissions could describe higher mass concentrations for nitrate and ammonium due to e.g., meteorological conditions and abundant precursors in winter time.

**R1C11:** L304-306, I agree with you that boundary layer mixing played a significant role to determine the measured org diurnal, but I don't think you can neglect the fact of evaporation of semi-volatile organic compounds in the middle day, especially in spring and summer seasons.

**AC11:** Thank you for the comment. We agree with the reviewer that the evaporation of semi-volatile particles is one of the reasons for the reduction of total OA during the day time (Schaap et al., 2004; Keck and Wittmaack, 2005). Therefore, we have added the following line to this section:

**Line 370-371 (revised version):** During warm days, evaporation of semi-volatile organics from the particle phase cannot be completely excluded (Schaap et al., 2004; Keck and Wittmaack, 2005).

**R1C12:** L331-332, Since you have observed very similar nighttime HOA concentrations in all seasons (Fig. 6), could the slightly different HOA concentration between summer and winter be explained by evaporation of HOA or photochemical conversion to LO/MO-OOA in summer?

**AC12:** Thank you for the comment. In general, low temperature in winter results in condensation of POA, which has been observed in all 22 sites in Chen et al. (2022). However, the temperature might play a role in the slightly lower HOA concentration during the night in summer time by emphasizing the

evaporation and oxidation processes of the emitted particles (Saha, et al., 2018). The difference between the HOA concentration during the night is very small, and most certainly can cover by the uncertainties of the PMF results (± 32.5 %, Fig. S2).

Moreover, we will not expect any photochemical aging during the night but the highest nighttime ozone concentration might also be considered as a potential aging mechanism (Kodros, et al., 2020).

The following sentences have been added to the HOA paragraph:

**Line 422-425 (revised version):** Nevertheless, the differences between HOA mass concentration during the night time from summer to winter season (Fig. 6) are small and can be covered by the uncertainties of PMF result (± 32.5 %, Fig. S2), however, it can be explained by different emission sources, condensation of POA (Chen et al., 2022), evaporation, oxidation processes (Saha, et al., 2018), and potential night time aging process by high ozone concentration (Kodros, et al., 2020).

**R1C13:** L335-337, HOA concentrations observed in summer and winter were very similar, (Fig. 6), sounding more like the characteristics of local emission sources. This is also in line with the usual view of HOA as a local emission factor. The city of Leipzig is located in the sector of HOA wind rose (Fig. 7). Considering that Leipzig is 50km away from to the measurement station, we assume that the average wind speed is 4m/s, and it took only 3 hours to transport Leipzig city aerosol particles to the measurement station. As you have already stated that HOA is mainly emitted by household heating and a minor source of traffic, is it possible that the city of Leipzig, as a local/or regional source, is a HOA contributor? Moreover, here you have attributed HOA to be a long-range transport one in winter. You have shown two peaks in the morning and evening in the diurnal cycle, which sounds they were more likely related to traffic rush hours, do you think if the long-range transported HOA can explain this typical diurnal cycle?

**AC13:** Thank you for highlighting these points. The NWR plot of HOA (Fig. 7) presents the surrounding/local and transported emissions and also two directions which can be associated with Leipzig (with approx. 600,000 inhabitants, in 50 km south-westerly sector of the measuring site) and Torgau (with approx. 20,000 inhabitants, in 7 km north-easterly sector) cities. Therefore:

- Surrounding/locally form HOA, is mostly related to household heating (van Pinxteren, et al., 2020) for all year, and partly related to traffic emission coming from Melpitz village itself and the main street (B 87, approx. 1.5 km north of the station), which could explain the two peaks linked to the traffic rush hours.
- However, these two sectors together with cluster and back-trajectory analysis (Fig. 9), can explain the long-range transported emissions arriving at Melpitz. Clusters CS-A1, CS-A2, and CS-C1 in the winter time, and cluster WS-A1 in summer (Fig. 9), present three different air masses coming to Melpitz which include part of the HOA.

Regarding the points mentioned above, the following sentences have been modified:

**Line 403-415 (revised version):** Analyzing the NWR plots demonstrates the highest HOA mass concentration was observed at low wind speed during the warm period (Fig. 7) indicating, rather local emission sources. While during the cold period a clear increase of the mass concentration can be associated with the highest wind speed (> 10 m/s) mostly coming from the North to East sector. During periods with wind speeds below 10 m/s, the two dominant wind sectors (NE and SW) can be observed. The first one might be associated with emission plumes coming from either the surrounding traffic emissions (the federal street B 87), as well as the domestic emissions are associated not with house heating in summer but with hot water production (van Pinxteren et al., 2020), as well as the city of Torgau (with approx. 20,000 inhabitants, distance from 7 km). Although the SW sector shows a lower HOA mass concentration in comparison to the NE one, it corresponds to the direction of the city of Leipzig (above 600 000 inhabitants, approx. 50 km). Therefore, it might be associated with the influence of the pollution plume of the city of Leipzig.

The diurnal patterns of HOA reproduced two peaks in the morning and evening for all seasons (Fig. 6), which is related to traffic rush hours and linked to surrounding emissions from the main street (B 87,

approx. 1.5 km north of the station), Melpitz village itself, and emissions coming from Leipzig and Torgau residential areas.

**R1C14:** L441-443, Though I agree to that night chemistry may play a role, how about the effect of boundary layer mixing on the diurnal cycle during the daytime?

**AC14:** Thank you for the comment. We agree with the reviewer that the boundary layer has an effect on the particles such as MO-OOA and the following sentence has been added to this part in the revised manuscript:

**Line 523-526 (revised version):** Meanwhile, MO-OOA diurnal cycles presented a seasonal variation as well, with a remarkable enhancement in the evening and night time during winter (Fig. 6), indicating a potential regional formation mechanism containing night time chemistry (Tiitta et al., 2016), and descending pattern from night time to day time due to planetary boundary layer effect.

**R1C15:** L445-447, Sulfate is also considered as a regional or long-range transported chemical species (e.g. Zhang et al., 2005). Does a better correlation between MO-OOA and sulfate indicate the nature of long-range transported MO-OOA in this study? (Zhang, Q., et al., Hydrocarbon-like and oxygenated organic aerosols in Pittsburgh: insights into sources and processes of organic aerosols, Atmos. Chem. Phys., 5, 3289-3311, 2005.)

**AC15:** We thank the reviewer for the comment. We agree with the reviewer that sulphate could both originate locally and be transferred from other sites to the measuring site, which we previously discussed sulphate in Sect. 3.1. Moreover, we also mentioned a good correlation between MO-OOA with sulphate has been found (Table. 1). Nevertheless, it is pretty unspecific to state the nature of MO-OOA, since MO-OOA (same as LO-OOA) could hardly be associated with a single chemical process nor aging of a specific source (like transported or locally formed SOA) because the exact chemical composition of OOA are unknown and the aging of the primary sources could also lead to OOA profiles (Jimenez, et al., 2009). Consequently, the correlation of MO-OOA and sulphate only indicates that both are secondary compounds without a clear indication of their primary sources (anthropogenic or biogenic).

**R1C16:** L454-456, are you saying nitrate was formed locally or via long-range transport?

**AC16:** Thank you for the comment. We are going to reply to this comment based on the discussion we have had on R1C1 and R1C8. This is true that nitrate has regional background sources in Melpitz (such as night time production, agricultural fertilization especially in spring, and traffic emission), but from the NWR plot together with cluster and back-trajectory analysis (Fig. 9) the long-range transported nitrate can be seen as well which are coming from different directions to measuring station.

**R1C17:** L458-459, As you have stated in lines 248-251 that nitrate is a long-range transported species, does a good correlation between LO-OOA and nitrate indicate that LO-OOA is also mainly from long-range transport?

**AC17:** We thank the reviewer for the comment. Regarding the discussion in R1C8 and R1C16, which are explaining the two different sources for nitrate as regional and transported are recognized in our study. The correlation of nitrate and LO-OOA is also discussed in lines 458-459 (preprint version) mostly for winter time, which can confirm the long-range transported origin of LO-OOA. Therefore, the following section has been revised.

**Line 540-542 (revised version):** Furthermore, since nitrate could be originated locally or arrived from a long distance to Melpitz (Sect. 3.1.1), with a good correlation between LO-OOA and nitrate ($R^2 = 0.59$) during winter, the long-range transported LO-OOA from different directions reaching to measuring site could be explained (Fig. 7).

**Technical corrections:**

**R1C18:** L23, the sentence "Melpitz represents due to its location the Central European aerosol" reads odd. Please reword it.

**AC18:** The text was revised as follows:

**Line 19-24 (revised version):** Here, the chemical composition and organic aerosol sources of submicron aerosol particles measured by an aerosol chemical speciation monitor (ACSM) and a multi-angle absorption photometer (MAAP) were investigated at Melpitz from September 2016 to August 2017. The location of the station at the frontier between Western and Eastern Europe makes it the ideal place to investigate the impact of long-range transport over Europe. Indeed, the station is under the influence of less polluted air masses from westerly directions and more polluted continental air masses from Eastern Europe.

**R1C19:** L123, please give a full name of the acronym when it appears for the first time.

**AC19:** We corrected the text and wrote the full name of PNSD as 'particle number size distribution' in line 139 of the revised version.

**R1C20:** L248, Fig. S3→Fig. 3?

**AC20:** Corrected. It is 'Fig. S8', line 307 revised version.

**R1C21:** L301, …in night time, …

**AC21:** Corrected: 'in night time', line 365 revised version.

**R1C22:** Fig.S2, the pixel resolution in Fig. S2 is rather poor. What does the radius axis in each rose plot represent?

**AC22:** Fig. S2 from pre-print version was replaced with high resolution Fig. S3 in the supplementary. The radius axis represents the wind speed in each plot.

**R1C23:** L372, Fig. 11?

**AC23:** We have removed 'Fig. 11' in the text since it was a mistake and there is no Fig. 11.

**R1C24:** L375, Fig. 4b→Fig.4c

**AC24:** Thank you for the point, however there is no Fig, 4c in the manuscript.

**R1C25:** L510, Fig. S3 and S4→Fig. S4 and S5?

**AC25:** The numbers of tables have been corrected: 'Tables S4 and S5', line 574 revised version.

**R1C26:** L579, Fig. 11?

**AC26:** The number of figures has been corrected as 'Fig. S4', line 648 revised version.

**References:**

[revised manuscript text omitted]

---

## Author Comment (AC2)

**Responses to the reviewer's comments on**
**"A one-year ACSM source analysis of organic aerosol particle contributions from anthropogenic sources after long-range transport at the TROPOS research station Melpitz" by Atabakhsh et al.**

Please find below the response to reviewer #2:
* * *
**Reviewer #2 (R2)**:

The authors thank the reviewer for providing constructive comments and insightful suggestions on the manuscript. We highly appreciate your time in reviewing the manuscript. The point-by-point response to all the comments and suggestions of reviewer #2 (R2) is provided in the following sections. For easy visualization, the reviewer's comments (R2 C) are provided in blue and the author's response (AR) is given in black color below the reviewer's comment. All the comments/suggestions were taken into consideration and incorporated in the revised manuscript which has improved the quality of the revised manuscript. The revised parts of the manuscript along with the references are shown in red.

Review on article "A one-year ACSM source analysis of organic aerosol particle contributions from anthropogenic sources after long-range transport at the TROPOS research station Melpitz" by Atabakhsh et al.

Article is based on year-long dataset where chemical composition is broadly characterized using ACSM and MAAP and source apportionment used to characterize the sources of PM. Article is easy to follow and understandable. However, the major issue seems to be that most of the data is already published by previous publications, at least Chen et al., 2022 (source apportionment at least for ACSM) and Poulain et al., 2020 (composition, source areas). If this is the case, it would be utmost important to firstly clarify and explain in experimental section explain the differences between these articles and in more detail give the novelty value of this dataset throughout the manuscript. It is clear that all results from 22 datasets are not covered by Chen et al, but at the moment without reading both side-by-side it is not clear what is. Also, the numbers (e.g. average mass and contributions) seem to be slightly different between this and Chen et al. article, which to me is not clear why. Was the data re-analyzed? In general, I think there are some nice results from the BC source apportionment and cluster analysis, but it is not exactly clear what is covered by previous articles, especially Poulain et al., 2020.

I propose re-writing/re-focusing this article so that it firstly identifies the novelty value and focuses on that more clearly. In this large and interesting dataset, this should not be a large issue and may not be that large of a work. Finally, the language should be checked thoroughly before resubmitting.

**AC:** We thank the reviewers' comments, however, we must emphasize some misunderstanding regarding the citing papers mentioned above. We agree that the source apportionment developed in this paper was already integrated into the European ACSM/AMS overview paper from Chen et al (2022). The main part of the Chen et al (2022) methodological paper was to establish the best practice for rolling windows PMF analysis. For this purpose, the results obtained on 22 different ACSMs over Europe were compared in a phenomenology discussion to first emphasize the robustness of the approach and second discuss the differences between the type of station (urban and rural-background) over Europe. However, Chen et al., (2022) did not discuss how the different sources were identified, nor depict the change of the entire chemical composition of the particles over the season (organic, inorganic, black carbon), nor the influence of the meteorological conditions at any station (especially the impact of air mass trajectory), nor the link between the organic aerosol and the black carbon to discuss the different source of black carbon, which all of them are the main focus of the present manuscript. Therefore, the present work provides valuable additional information compared to Chen et al., 2022, which is essential to better understand the complexity of the air mass dependency on air quality in central Europe.

Moreover, we think that the reviewer mixed up two of our previous papers. The cited paper Poulain et al. (2020, published in AMT) was focused on the quality assurance of the ACSM measurements made at Melpitz. In that paper, the ACSM data were systematically compared with collocated measurements including MPSS, off-line $PM_1$, and $PM_{2.5}$ high-volume filter samples for water-soluble ions (sulphate,

nitrate, and ammonium) as well as OC/EC. In Poulain et al. (2020) paper, neither source apportionment nor wind direction or air mass trajectory analysis was performed. In another paper from Poulain et al. (2021, ACP special issue HCCT-2010), source apportionment analysis on AMS data was published as well as some air mass cluster analysis. However, the measurements were not performed at Melpitz station. It is possible that the reviewer confuses the three following papers:

- Poulain et al., 2020: https://doi.org/10.5194/amt-13-4973-2020, The ACSM robustness, quality assurance, and impact of upper size cutoff diameter have been discussed at the same station, Melpitz.
- Poulain et al., 2021: https://doi.org/10.5194/acp-21-3667-2021, The sources, and impact of long-range transport on carbonaceous aerosol have been discussed in this study for Central Germany during HCCT-2010.

We appreciate reviewer spotted the difference, between the current study and Chen et al., (2022), The BC data were accidentally applied CDCE in Chen et al. (2022), which made it around 2 factors higher than current paper. Therefore, it causes the differences in fractions of $PM_1$ compositions. The co-authors of this paper (also the first author and corresponding author of Chen et al. (2022)) will try to correct this mistake soon.

**Detailed comments:**

**R2C1:** Abstract: lines 16-25, maybe consider condensing this text. It repeats some things several times.

**AC1:** Thank you for asking to condense this text. Lines 16-25 of the preprint version are rewritten as follows:

**Line 16-21 (revised version):** To better understand their sources, investigations have been focused on urban areas in the past, while rural-background stations are normally less impacted by surrounding anthropogenic sources. Therefore, they are predisposed for studying the impact of long-range transport of anthropogenic aerosols. Here, the chemical composition and organic aerosol sources of submicron aerosol particles measured by an aerosol chemical speciation monitor (ACSM) and a multi-angle absorption photometer (MAAP) were investigated at Melpitz from September 2016 to August 2017.

**R2C2:** Line 47: PMs -> PM fractions

**AC2:** Since we are using 'fractions' as 'contribution' during the whole manuscript, we decided to use 'PMs' which means all the particulate matters, to prevent confusion later on. However, the following part of manuscript have been revised:

**Line 45-49 (revised version):** The submicronic particles known as $PM_1$ (particles with an aerodynamic diameter less than 1 µm), not only have a negative impact on human health (Pop and Dockery, 2016; Daellenbach et al., 2020) but also have a significant effect on visibility (Shi et al., 2014) and climate (Shrivastava et al., 2017). It is ability to penetrate to respiratory system make it more dangerous, therefore more relevant to mitigate adverse health impact.

**R2C3:** Line 48: please consider adding another reference to epidemiological study that links PM and health more broadly. Several articles exist (e.g. Pope, C. A. and Dockery, D. W.: Health Effects of Fine Particulate Air Pollution: Lines that Connect, J. Air & Waste Manage. Assoc 56, 35, 2006.)

**AC3:** Thank you for the comment and suggestion. We agree with the reviewer which PM health effect is not negligible. Therefore, we added the suggested reference to this statement: 'Pop and Dockery, 2016', line 47 revised version.

**R2C4:** Line 63: was -> has been

**AC4:** Thank you for the correction. The verb 'has been', line 63 revised version.

**R2C5:** Lines 85-90: As the source apportionment is published already by Chen et al., article, I would recommend re-defining this scope. As Chen et al. had 22 datasets, it is likely that not many details were given for each site, but the basic results of source apportionment e.g. timeseries of factors were given there. The novelty value is not clear.

**AC5:** Thank you for the comment and recommendation. The overview paper by Chen et al., (2022) focused on, first: the determination of a unified rolling PMF analysis approach and its validation at different sampling sites; and second: the comparison of the absolute sources at the different European sites. In the present work, we considered the entire aerosol chemical composition not only the organic fraction, with a specific focus on the influence of the wind direction and air mass on the chemical composition. Furthermore, Chen et al., (2022) only focused on the identification of the organic sources at each sampling site, while our study also investigated the sources of eBC as well with three anthropogenic sources related to eBC. This last approach is important since the currently used aethalometer model is limited to splitting the eBC in contributions of fossil fuel ($eBC_{ff}$) and wood burning ($eBC_{wb}$). Taking it all together, we agree that the final results of the organic PMF analysis were already presented in Chen et al (2022). But the present manuscript provides a significant additional and detailed discussion on the organic sources as well as new results to make it a stand-alone work without repeating the result discussed in the previous work.

The following text is revised in the introduction section for the mentioned paragraph in order to clarify the novelty of this study in comparison to Chen et al., 2022 and other publications as well:

**Line 84-95 (revised version):** The current study comprehensively investigates the $PM_1$ aerosol particle chemical compositions and the various OA sources for Melpitz as a rural-background station, based on ACSM and multi-angle absorption photometer (MAAP) measurements from September 2016 to August 2017, using the most advanced rolling PMF with ME-2 implemented in the SoFi Pro package (Datalystica Ltd., Villigen, Switzerland) (Parworth et al., 2015; Canonaco et al., 2013; Canonaco et al., 2020). Although previous papers already considered this dataset, they were focused on quality assurance (Poulain et al., 2020) to depict the European aerosol chemical composition (Bressi et al., 2021 and Chen et al., 2022) or the relationship between the CCN properties (Wang et al., 2022, Schmale et al., 2017), none of these papers were focused on carbonaceous source identification (OA and eBC) nor discussed the strong dependency of the aerosol chemical composition to the air mass origin. Therefore, a multi-linear regression model was used to estimate the contribution of equivalent black carbon (eBC) to the various primary organic PMF factors such as hydrocarbon-like organic aerosol, biomass burning organic aerosol, and coal combustion organic aerosol. Meanwhile, to better understand the emission area of $PM_1$ chemical composition and PMF factors, the influence of air mass origin was investigated based on self-developed back-trajectory cluster methods (BCLM).

**R2C6:** Chapter 2. experimental. Clearly explain what is previously published and what is done first time in this article so reader can understand the novelty value of this article.

**AC6:** Thank you for the comment. The previous paper (Chen et al., 2022) presented the developed rolling PMF model by validating it on the various data from different stations and also discussed the absolute sources of the organic and inorganic stations with no regard for geographical origin. While the scope of current study is to provide detailed descriptions of chemical composition and OA sources with additional evidences of long-range transportation airmasses, also the $eBC$-$PM_1$ fraction was discussed.

The following text is revised in the introduction section for the mentioned paragraph in order to clarify the novelty of this study in comparison to Chen et al., 2022 and other publications as well:

**Line 84-95 (revised version):** The current study comprehensively investigates the $PM_1$ aerosol particle chemical compositions and the various OA sources for Melpitz as a rural-background station, based on ACSM and multi-angle absorption photometer (MAAP) measurements from September 2016 to August 2017, using the most advanced rolling PMF with ME-2 implemented in the SoFi Pro package (Datalystica Ltd., Villigen, Switzerland) (Parworth et al., 2015; Canonaco et al., 2013; Canonaco et al., 2020). Although previous papers already considered this dataset, they were focused on quality assurance (Poulain et al., 2020) to depict the European aerosol chemical composition (Bressi et al., 2021 and Chen et al., 2022) or the relationship between the CCN properties (Wang et al., 2022, Schmale et al., 2017), none of these papers were focused on carbonaceous source identification (OA and eBC) nor discussed

the strong dependency of the aerosol chemical composition to the air mass origin. Therefore, a multi-linear regression model was used to estimate the contribution of equivalent black carbon (eBC) to the various primary organic PMF factors such as hydrocarbon-like organic aerosol, biomass burning organic aerosol, and coal combustion organic aerosol. Meanwhile, to better understand the emission area of $PM_1$ chemical composition and PMF factors, the influence of air mass origin was investigated based on self-developed back-trajectory cluster methods (BCLM).

**R2C7:** Line 121: Please explain the conversion, not everybody have time to check the reference "Conversion of the eBC mass concentration from the PM10 inlet to the ACSM PM1 cut-off was made by applying a correction factor of 0.9 following Poulain et al (2011)." and add discussion why the eBC measured for PM10 is shown in plots with ACSM that measures mainly submicron particles. Also, to the ACSM chapter would be good to add the size range of particles that ACSM measures.

**AC7:** We apologized for the confusion. It is true that the MAAP is measuring the $PM_{10}$ eBC, while the ACSM has a near $PM_1$ cut-off. Therefore, soot concentration for $PM_1$ is required to perform a proper $PM_1$ mass closure. To better understand the split of the eBC between $PM_1$ and $PM_{10}$, temporary but parallel measurements using two MAAPs, one connected to the $PM_{10}$ inlet and a second one to a $PM_1$ inlet were performed at Melpitz. This comparison demonstrated that the soot concentration in $PM_1$ is around 90 % of that in $PM_{10}$, with this ratio being only weakly time-dependent. Consequently, for this study, we estimated soot in the $PM_1$ by multiplying the soot concentration on $PM_{10}$ by a constant factor of 0.90. In the entire manuscript, the reported eBC corresponds to the estimated eBC in the $PM_1$ range. To avoid further confusion the text was changed as follows:

**Line 135-138 (revised version):** The eBC mass concentration from the $PM_{10}$ data was multiplied by a constant factor of 0.9 following Poulain et al (2011) to estimate the eBC mass concentration in the $PM_1$ fraction. Consequently, all the eBC mass concentrations reported and discussed here correspond to the eBC in the $PM_1$ fraction and are referred to as eBC-$PM_1$.

**R2C8:** Line 129: very long sentence. also, the sentence speaks both about levoglucosan and monosaccharide anhydrides in plural. maybe clarify if you measured only levo or also galactosan and mannosan.

**AC8:** Thank you for the comment. All the other sugars such as galactosan and mannosan are measured by HPAEC-PAD, but since we only used levoglucosan in our study, we just kept levoglucosan in this part.

**Line 146-148 (revised version):** Levoglucosan as a tracer for wood burning combustion was measured following Iinuma et al., (2009) using high performance anion exchange chromatography coupled with an electrochemical detector (HPAEC-PAD).

**R2C9:** Line 139-140: please give the version numbers for both SOFI-pro and igor. in future, it usually helps to know which version was used, because they tend to improve and usually some bugs are fixed.

**AC9:** Thank you for the suggestion and points. We have included the number of versions for both SoFi and Igor Pro in the sentences as follows:

**Line 154-156 (revised version):** The PMF method was used to allocate the source of the OA (Paatero and Tappert, 1994) through the Source Finder professional (SoFi Pro, version 8.0.3.1, Canonaco et al., 2021) software package (Datalystica Ltd., Villigen, Switzerland), within the Igor Pro software environment (Igor Pro, version 8.04, Wavemetrics, Inc., Lake Oswego, OR, USA).

**R2C10:** Line 206: maybe compare also to long-term dataset presented by Poulain et al., 2020?

**AC10:** Thank you for the comment. As the referee suggested, we made a comparison between our study and Poulain et al., (2020) with data period from June 2012 to November 2017 and similar values for mean mass concentration and seasonal trend could be observed. The following line has been added to the mentioned section:

**Line 248-253 (revised version):** Compared to previous ACSM long-term measurements of Poulain et al., (2021) at the same station, a similar mean mass concentration of $PM_1$ was observed in the period from June 2012 to November 2017 (Poulain et al., 2021: 10.23 µg/m$^3$ and this study: 10.49 µg/m$^3$; respectively), and presented same seasonal trends for all the chemical species (Table. S2) with a highest mass concentration in the winter and lowest mass concentration in the summer time (13.15 µg/m$^3$ and 7.64 µg/m$^3$, respectively; Table. S2). Consequently, the results obtained from the current study can be considered as a representative ACSM study for Melpitz station.

We also added the Table. S2 to the Supplementary file:

**Table. S2: PM$_1$ seasonal mass concentration (µg/m$^3$) of Poulain et al, (2021), and average from the current study.**

| Species | Fall | Winter | Summer | Spring | Average | the current study |
|---------|------|--------|--------|--------|---------|-------------------|
| Org | 3.83 | 4.58 | 4.41 | 4.28 | 4.27 | 4.84 |
| SO$_4$$^{2-}$ | 1.53 | 1.86 | 1.37 | 1.41 | 1.54 | 1.67 |
| NO3- | 2.24 | 3.79 | 0.90 | 3.07 | 2.50 | 2.16 |
| NH$_4$$^+$ | 1.10 | 1.63 | 0.65 | 1.35 | 1.18 | 1.11 |
| Cl- | 0.04 | 0.07 | 0.01 | 0.05 | 0.04 | 0.05 |
| eBC-PM$_1$ | 0.69 | 1.22 | 0.30 | 0.56 | 0.69 | 0.66 |
| Tot | 9.43 | 13.15 | 7.64 | 10.72 | 10.23 | 10.49 |

**R2C11:** Line 224-227. Please clarify what this means and why these measurement periods are important?

**AC11:** Thank you for the comment. The discussed lines are: "Moreover, with enhanced irradiations in summer, sulphate formation from photochemistry could be enhanced as well. This result is consistent with the mean $PM_1$ mass concentration measured by AMS for the three periods during fall (16. September.2008 to 03. November.2008), winter (24. February.2009 to 25. March.2009), and summer (23. May.2009 to 09. June.2009) campaigns reported by Poulain et al., (2011)." in the preprint version. We are going to answer this comment in two parts:

- Part one: Since the photochemical oxidation process of sulphur dioxide is direct cause of the increment of sulphate in the atmosphere, therefore sulphate can increase during the summer due to the high solar radiation.
- Part two: To make a comparison of the Melpitz study between the present study with an AMS study from Poulain et al., (2011), the three periods have been discussed since the measurement campaign of Poulain et al., (2011) took place on these three limited periods which covered different parts of the seasonal cycle, we also took the same period of our data, and compared them. It showed that the sulphate contribution to total $PM_1$ is higher in the summer time rather than in winter time due to the photochemical oxidation process.

Furthermore, we made a comparison between our study and Poulain et al., (2020) with a long-term ACSM dataset (from June 2012 to November 2017) and similar values for sulphate mean mass concentration could be observed, (15 % in Poulain et al., and 16 % in the current study).

To avoid the confusion, we have revised the discussed section and the following line has been added to the mentioned section:

**Line 274-282 (revised version):** Moreover, the sulphate contribution to the total $PM_1$ was higher during the summer than winter time, since with enhanced irradiations in summer, sulphate formation from photochemistry could be enhanced as well. This sulphate higher contribution in summer over winter is consistent with the mean $PM_1$ mass concentration measured by AMS for the three periods during fall (16. September.2008 to 03. November.2008), winter (24. February.2009 to 25. March.2009), and summer (23. May.2009 to 09. June.2009) campaigns reported by Poulain et al., (2011). In comparison with previous ACSM long-term measurements of Poulain et al., (2021) at Melpitz station, a similar mean mass concentration of sulphate was observed in the period from June 2012 to November 2017 (Poulain et al., 2021: 1.54 µg/m$^3$ and this study: 1.67 µg/m$^3$; respectively; Table. S2). This comparison indicates the current study as a case study of ACSM for Melpitz station within 5-year ACSM data, with the best data coverage of time in a year.

**R2C12:** Line 209-211: "Fig. S2 presents the coming high polluted air masses for total PM₁ to the measurement site in the current study; the polluted Eastern Europe flow with high mass concentration and south-west with low mass concentration was more clearly found in winter time rather than in other seasons, which will be comprehensively discussed in the Sect. 3.4." should it be e.g. "lower mass concentration for SW", as the lowest were the for north west and south.. I am struggling to understand where this is compared.

**AC12:** Thank you for the comment. We compared the result of the Melpitz station with other rural-background stations from two other publications (Bressi et al., 2021 and Chen et al., 2022) throughout Europe, lines 257 and 261 of the revised version. From the comparison, we found that there is a similarity in annual PM₁ mean mass concentration between Melpitz and other rural-background stations in mid-latitude Europe.

We also used 'lower' instead of 'low' in line 254 revised version.

**R2C13:** Line 212: maybe here also comparison to long-term concentrations shown by Poulain et al., 2020? this would give reader an idea whether this chosen year was typical or an anomaly.

**AC13:** Thank you for the comment. As the referee suggested, we made a comparison between our study and Poulain et al., (2020) with data period from June 2012 to November 2017 and similar values for mean mass concentration and seasonal trend could be observed. The following lines have been added to the mentioned section:

**Line 248-253 (revised version):** Compared to previous ACSM long-term measurements of Poulain et al., (2021) at the same station, a similar mean mass concentration of PM₁ was observed in the period from June 2012 to November 2017 (Poulain et al., 2021: 10.23 µg/m³ and this study: 10.49 µg/m³; respectively), and presented same seasonal trends for all the chemical species (Table. S2) with a highest mass concentration in the winter and lowest mass concentration in the summer time (13.15 µg/m³ and 7.64 µg/m³, respectively; Table. S2). Consequently, the results obtained from the current study can be considered as a representative ACSM study for Melpitz station.

We also added the Table. S2 to the Supplementary file:

**Table. S2: PM₁ seasonal mass concentration (µg/m³) of Poulain et al, (2021), and average from the current study.**

| Species | Fall | Winter | Summer | Spring | Average | the current study |
|---------|------|--------|--------|--------|---------|-------------------|
| Org | 3.83 | 4.58 | 4.41 | 4.28 | 4.27 | 4.84 |
| SO₄²⁻ | 1.53 | 1.86 | 1.37 | 1.41 | 1.54 | 1.67 |
| NO3- | 2.24 | 3.79 | 0.90 | 3.07 | 2.50 | 2.16 |
| NH₄⁺ | 1.10 | 1.63 | 0.65 | 1.35 | 1.18 | 1.11 |
| Cl- | 0.04 | 0.07 | 0.01 | 0.05 | 0.04 | 0.05 |
| eBC-PM₁ | 0.69 | 1.22 | 0.30 | 0.56 | 0.69 | 0.66 |
| Tot | 9.43 | 13.15 | 7.64 | 10.72 | 10.23 | 10.49 |

**R2C14:** Line 232. clarify how do you separate locally formed sulphate from any other source?

**AC14:** We thank the reviewer for the comment. The non-parametric wind regressions model (NWR) had been used to investigate not only the local but also the transported emission sources (Marin, et al., 2019) in Sect. 3.1-3.3. With this regard, if high concentrations are observed at low wind speeds, they are recognized as local sources, whereas high concentrations at high wind speeds are more transported ones; which means: when there is almost no wind, the concentration is local and spread throughout the station, whereas when there is more wind, the concentration is systematically associated with a specific wind sector, which may correspond to the transport process. In order to clarify the statement, the following lines have been added to the NWR definition:

**Line 214-216 (revised version):** Non-parametric wind regressions (NWR) were used to approximate the OA source concentrations at a given wind direction and speed (Henry et al., 2009) in order to investigate not only the local but also the prevalent wind sector associated with transported emission sources (Marin, et al., 2019).

Since the wind direction did not provide any information on the air mass origin, cluster analysis on air mass back-trajectories was used to identify the long-range transported emissions and it was discussed in Sect. 3.4. However, we briefly included them in Sect. 3.1-3.3, we decided to keep Sect. 3.4 (Seasonal air mass clustering) as an overview of this study which comprehensively discussed the origins of chemical species and PMF sources. Therefore, some parts of Sect. 3.1-3.3 have been revised as follows:

**Line 286-291 (revised version):** Furthermore, the NWR plots (Fig. S3) show that during the winter time, sulphate mostly comes from the north and east sectors with wind speeds above 5 m/s which can be associated with dominant transported sulphate sources. Although the eastern wind sector remains visible for the sulphate in the summer time, the high concentrations of sulphate can be observed during periods with low wind speed and without a specific wind sector; which corresponds to local sulphate formation. Sect. 3.4 will go into detail about the long-range transported emissions later on.

**R2C15:** Line 258: what is the detection limit for chloride in ACSM? is 0.05 µg m-3 above dl?

**AC15:** The mean mass concentration for chloride in this study is 0.05 µg/m$^3$, which is above the chloride detection limit, < 0.011 µg/m$^3$ is the detection limit for chloride (Ng et al., 2011).

**R2 C16:** Chapter 3. should the name be results and discussion?

**AC16:** Thank you for the suggestion. The title of Sect. 3 has been changed to 'Results and discussion' as suggested, in line 236 (revised version).

**R2C17:** For chapters 3.1.1 and 3.1.2. I am bit struggling to see the novelty value. Please highlight the new results and their meaning/impact for science renewal.

**AC17:** Thank you for the comment. The mentioned Sect. (3.1.1 and 3.1.2) are discussions around ACSM and MAAP data analysis. The seasonality of chemical compositions in individual sites were not really covered in Chen et al. (2022). Also, the long-range transportation influence to this specific site was not investigated by Chen et al. (2022), therefore, giving the special location of Melpitz, the discussion in 3.1.1 and 3.1.2 are important to address these questions. In order to highlight the novelty of this manuscript, we have revised the following lines in the introduction section:

**Line 84-95 (revised version):** The current study comprehensively investigates the PM$_1$ aerosol particle chemical compositions and the various OA sources for Melpitz as a rural-background station, based on ACSM and multi-angle absorption photometer (MAAP) measurements from September 2016 to August 2017, using the most advanced rolling PMF with ME-2 implemented in the SoFi Pro package (Datalystica Ltd., Villigen, Switzerland) (Parworth et al., 2015; Canonaco et al., 2013; Canonaco et al., 2020). Although previous papers already considered this dataset, they were focused on quality assurance (Poulain et al., 2020) to depict the European aerosol chemical composition (Bressi et al., 2021 and Chen et al., 2022) or the relationship between the CCN properties (Wang et al., 2022, Schmale et al., 2017), none of these papers were focused on carbonaceous source identification (OA and eBC) nor discussed the strong dependency of the aerosol chemical composition to the air mass origin. Therefore, a multi-linear regression model was used to estimate the contribution of equivalent black carbon (eBC) to the various primary organic PMF factors such as hydrocarbon-like organic aerosol, biomass burning organic aerosol, and coal combustion organic aerosol. Meanwhile, to better understand the emission area of PM$_1$ chemical composition and PMF factors, the influence of air mass origin was investigated based on self-developed back-trajectory cluster methods (BCLM).

**R2C18:** For the chapter 3.1.1.: please add other references also, outside Melpitz.

**AC18:** Thank you for the comment and suggestion. As the referee suggested, we compared the present study with other rural-background stations from Bressi et al., (2021) which only presented the winter and summer results. From the comparison of diurnal profiles between two studies for Melpitz station (Fig. S4 from Bressi et al., 2021 and Fig. 3 from our study) with different time coverage, all the main chemical compositions (named below) showing the similar patterns:

- Sulphate presents an increase during the day and decreased in the night time for summer the season, with no significant changes in the winter season.
- Nitrate and ammonium present a decreasing pattern during the day, and an increasing pattern during the night time in both summer and winter time.
- Organic also shows similar patterns, decreasing during the day and increasing during the night in both summer and winter time.

[Figure]

**Figure S4 from Bressi et al., (2021): daily cycles (median values) in the 4 main components of NR-PM$_1$ concentrations (µg/m$^3$) measured at the 21 selected sites in winter (DJF) and summer (JJA).**

[Figure]

[Figure]

**Fig. 3: Seasonal diurnal cycle of PM$_1$ for ACSM organic and inorganic species (Time is in UTC).**

In comparison with other rural-background stations, there is no similarity of the main chemical composition in each diurnal profile of the different stations, because every station has particular conditions such as geographical location, different types of sources, and various meteorological conditions.

We have added other stations to make a better comparison between rural-background stations in Europe as follow in Sect. 3.1.1 and 3.1.2:

**Line 265-270 (revised version):** Moreover, the comparison between Bressi et al., (2021) and current study (Fig. S4 from Bressi et al 2021, Fig. 3 from the current study) for Melpitz station with different time coverage shows that the daily variation of ACSM sulphate, nitrate, and ammonium are similar in both winter and summer seasons. In comparison with other ACSM/AMS rural-background stations in Europe (Fig. S4, Bressi et al., 2021), the mean daily cycle of the PM$_1$ chemical components (sulphate, nitrate, and ammonium) does not show a similar pattern to the other stations (Bressi et al., 2021) due to the different geographical location and meteorological conditions.

In addition to Sect. 3.1.1, we also added a similar comparison for the organic part in Sect. 3.1.2 as follows:

**Line 371-374 (revised version):** In comparison between Bressi et al., (2021) and the current study for Melpitz station, the daily variation of organic are similar in both winter and summer seasons, while there are differences between Melpitz with other rural-background stations due to the different geographical location and meteorological conditions (Bressi et al., 2021).

**R2C19:** Chapter 3.2. Source apportionment. I think quite many of the things discussed here (like factor identification) should go to experimental or supplementary material. These are fairly well known and are not results. I am struggling to find the novelty value of this chapter. The source apportionment is extensively discussed in literature, and results seem to be in-line with previous research results. Please, try to highlight the novelty further.

**AC19:** Thank you for the comment. Sect. 3.2 which is named 'source apportionment of OA', explains each of the organic aerosol sources by starting with a 2-3 lines definition of them. This is true that factor identification is already known and discussed in previous literatures, but we are explaining every PMF

factor as a result of this study by a definition of them which are not only strongly linked to the following statements, such as correlation with specific tracer, but in a way that it can also make it understandable for other readers outside of PMF/AMS-ACSM field as well. Furthermore, for some of them like OOAs, it is important to define the *m/z* 44 and 43, since the discussion cannot be completed without mentioning the *m/z* at the beginning of result discussion, and triangle plots could be discussed without these definitions. With respect to the reviewers' comment, we think that with 2-3 sentences about the factor identification, we can explain the PMF factors results with such details in a more reliable way.

**R2C20:** Line 359:" The high value of BBOA" I would rephrase to be "high concentration. "

**AC20:** Thank you for the suggestion. As suggested, the term 'high value' is changed to 'high mass concentration', line 436 (revised version).

**R2C21:** Chapter 3.3. Very interesting approach. I suggest moving the method description and formulas to the experimental and leaving here results and discussion. Please, try to discuss the value of results here also.

**AC21:** We thank the reviewer for the comment and suggestion. We moved the method description and formulas of eBC-PM$_1$ source apportionment to the experimental section (lines 195-212 revised version) under the title of '2.5 eBC-PM$_1$ source apportionment', and we left only its result in the 'Result and discussion' as suggested by reviewer: 3.3 Source apportionment of eBC-PM$_1$, lines 550-566 revised version.

Here by the multilinear regression model, we identified the contribution of primary organic aerosols named HOA, BBOA, and CCOA, on eBC-PM$_1$ for two reasons:

- First, previous literatures were not investigating the sources and emission area of carbonaceous aerosol based on OA-PMF sources,
- Second, eBC can be split in eBC$_{Cff}$ and eBC$_{wb}$ based on aethalometer model which means it can only consider there two sources, and no more.

To this order, we added the following paragraph to the manuscript.

**Line 91-94 (revised version):** Therefore, a multi-linear regression model was used to estimate the contribution of equivalent black carbon (eBC-PM$_1$) to the various primary organic PMF factors such as hydrocarbon-like organic aerosol, biomass burning organic aerosol, and coal combustion organic aerosol.

**R2C22:** Line 483: "CCOA appeared to have the largest source of eBC," I would suggest rephrasing to be CCOA appeared **to be** the largest source of eBC",

**AC22:** Thank you for the suggestion. As suggested, the term 'to have' is changed to 'to be' in the text, line 552 (revised version).

**R2C23:** Chapter 3.4 the titles could be more informative.

**AC23:** Thank you for the comment. The title of Sect. 3.4 has been changed from 'Impact of air mass origin and trajectory analysis' to 'Seasonal air mass clustering', Line 567 (revised version).

**R2C24:** Chapter 3.4. also, the technical description of cluster analysis and identified clusters would be more suitable for experimental than results. Also, please give details of the analysis, based on what the clusters were identified and how long data was used. figure legend says 13-years, text is speaking of a year.

**AC24:** Thank you for the comment. As the reviewer suggested, the technical description of cluster analysis has been moved to the experimental section (lines 229-235, 2.6 Air mass trajectory analysis) which is included detailed information on this technique. We apologize for the confusion about the Fig. 9 legend and manuscript text, which is corrected as follows:

**Line 1198 (revised version):** a) air mass classification based on one-year backward trajectories cluster analysis at 12:00 UTC.

**Now in Sect. 2.6, line 229-235 (revised version**): In this method, the different clusters can be divided according to the different seasons (CS: cold season; TS: transition season; and WS: warm season), and meteorological synoptic patterns (ST: stagnant; A1: anticyclonic with air mass coming from Eastern Europe; A2: anticyclonic with air mass coming from the west; C1: cyclonic with air mass coming from relatively south; C2: cyclonic with air mass coming from the west and south west). However, the clustering approach did not consider spring and fall separately, and therefore the transition clusters correspond to both spring and fall. Finally, a total of fifteen clusters were identified, corresponding to different meteorological conditions over the course of the year. Descriptive analysis, cluster processing, and data processes and products are all described in detail by Sun et al., (2020) and Ma et al., (2014).

**Now in Sect.3.4, line 568-574 (revised version):** As mentioned before, the geographical origin of the $PM_1$ chemical species and also PMF components are not only emitted from the surrounding area but also transported. Therefore, to better identify the origin of their sources, trajectory analysis, and their clustering analysis were applied using the self-developed back-trajectory cluster method (BCLM) (Sun et al., 2020; Ma et al., 2014; Hussein et al., 2006). Regarding this cluster approach, six air masses were identified at Melpitz station for the winter season, four air masses for the transition seasons, and five air masses for the summer season (Fig. 9a). The number of clusters with their corresponding mean mass concentration of $PM_1$ chemical species and PMF factors of organics are summarized in Table. 2 and with more details in Tables S4 and S5.

**R2C25:** Line 519: clarify what means "surrounding emission origin". is it like very local, regional or even LRT?

**AC25:** Thank you for the comment. 'Surrounding' corresponds to the local emissions, which means emissions from Melpitz village itself, around the measuring station as well as short-distance transport (that can be emissions from Leipzig and Torgau cities).

Furthermore, during winter time, one of the identified air masses from the BCLM model for Melpitz station which is recognized based on the synoptic patterns is CS-ST (which means cold season-stagnant), presenting the stable air condition around the station. In this situation, air masses can be known as surrounding emissions, and when we want to point to their origin, it has been called surrounding emission origin. It is similar to WC-ST air mass which means warm season-stagnant. To better understated and avoid the complexity and confusion in this part, we have revised the following statement:

**Line 579-580 (revised version):** These surrounding emissions refer to the emissions from Melpitz station directly, Melpitz village, and short distance transported particles like particles from Leipzig and Torgau.

**R2C26:** Line 516-517: "This cluster with the highest mass concentration of LO-OOA to the PM mass (2.73 μg/m3) could confirm the role of freshly formed SOA originating around the station from primary biomass burning and coal combustion emissions (mass concentration of 0.97 μg/m3 and 1.89 μg/m3, respectively)." please clarify this conclusion. Also, it is surprising that this happens in winter when UV is lowest. Did you see same in summer?

**AC26:** Thank you for the comment. SOA is considered to be formed by biomass burning and coal combustion as well, especially during the winter time when biogenic emissions are minimal (Lanz et al., 2010). In fact, the Aqueous-phase processing of biomass-burning emissions contributes to SOA formation, which this aqueous SOA absorbs both UV and visible light more efficiently that other OA components (Gilardoni et al., 2016). Furthermore, under dark conditions, biomass burning emissions age rapidly in the presence of $NO_2$ and $O_3$, producing a similar amount of SOA as under photochemical conditions (Kodros, et al., 2020). Therefore, with a high value of ozone during night time (Fig. S4), the dark aging chemistry can cause SOA formation in the winter season.

In order to clarify the mentioned statement, we have revised this section as follows:

**Line 580-585 (revised version):** This cluster presented the highest mass concentration of LO-OOA to the PM mass (2.73 μg/m$^3$). In fact, SOA is considered to be formed by biomass burning as well as coal combustion, particularly during the winter when biogenic emissions and UV radiation are low (Lanz et al., 2010; Kodros, et al., 2020). In this condition and in the presence of $NO_2$ and $O_3$, the biomass burning emissions could age rapidly and produce SOA. In conclusion, this cluster could confirm the role of freshly formed SOA which originated from the primary biomass burning and coal combustion emission (mass concentrations of 0.97 μg/m$^3$ and 1.89 μg/m$^3$, respectively).

**R2C27:** Line 617: please add better compared to what? also, add how you observed the improvement

**AC27:** Thank you for the comment. The mentioned statement is pointing to the advantage of the rolling approach compare to the regular PMF, which is performed over the entire dataset with the assumption that the OA profiles are static which can result in high errors for a long-term dataset because OA chemical fingerprints are expected to change over the time. Instead, in the rolling technique a small-time window moves in daily steps across the whole dataset instead of running PMF on the entire dataset. Therefore, the model can slowly adjust the factor profiles over various periods, which provides well-separate OA factors.

**Line 687-690 (revised version):** For OA source apportionment, PMF in a rolling fashion has been applied using the SoFi Pro, which provided the decomposition of time-dependent factor profiles that were able to better capture the variability of OA sources across seasons in comparison with the conventional seasonal PMF.

**R2C28:** Conclusions: please try to highlight the novelty of the results

**AC28:** We thank the reviewer for asking to highlight the novelty of the results, the new findings of the study are summarized as follows.

i.   In addition to OA source apportionment, the eBC-PM$_1$ source apportionment has been studied using a multilinear regression model based on primary organic aerosol sources; HOA, BBOA, CCOA. While it is common to determine the contribution of eBC$_{Cff}$ and eBC$_{wb}$ based on aethalometer model which means it can only consider these two sources, and no more.

ii.  There have been numerous PMF-based studies in the subject area of OA source apportionment. However, data coverage for most of them is for one year or less and also not for a one-year long continuous including four seasons.

iii. Since most of the previous source apportionment publications studied the urban and urban-background sites, there is still a lack of information about the sources in rural and rural-background sites which can better present the impact of long-range transport of anthropogenic aerosols.

iv.  Chen et al. (2022) provided a large overview of the European geographical OA sources but did not discuss the factors influence/control the OA sources at each site, while in the present study, the geographical and influence factor of OA sources has been investigated with NWR model, cluster and back-trajectory analysis.

v.   The study expresses the geographical origin of the PM$_1$ in terms of the origin of chemical composition, and the origin of PMF sources for a rural site. Using the cluster and back-trajectory analysis, the origin of sources results in terms of the different seasons and meteorology conditions.

vi.  Various source apportionment analysis has been done in the previous studies over the OA with the PMF approach. From Chen et al. (2022), only two stations reported CCOA and the second one is an urban station with coal combustion sources nearby. While this study investigated the existence of a coal combustion source over the year on Melpitz as a rural site. The cluster and back trajectory analysis are providing the geographical origin of this source for various seasons.

Conclusion: The following lines/paragraphs have been added/modified/rewritten in Sect. 4 "Conclusion" of the revised manuscript to incorporate the suggestions of the reviewer:

[revised manuscript text omitted]